# Brain Diffusion for Visual Exploration: Cortical Discovery using Large Scale Generative Models

**Andrew F. Luo**
Carnegie Mellon University
afluo@cmu.edu

**Margaret M. Henderson**
Carnegie Mellon University
mmhender@cmu.edu

**Leila Wehbe**\*
Carnegie Mellon University
lwehbe@cmu.edu

**Michael J. Tarr**\*
Carnegie Mellon University
michaeltarr@cmu.edu

## Abstract

A long standing goal in neuroscience has been to elucidate the functional organization of the brain. Within higher visual cortex, functional accounts have remained relatively coarse, focusing on regions of interest (ROIs) and taking the form of selectivity for broad categories such as faces, places, bodies, food, or words. Because the identification of such ROIs has typically relied on manually assembled stimulus sets consisting of isolated objects in non-ecological contexts, exploring functional organization without robust *a priori* hypotheses has been challenging. To overcome these limitations, we introduce a data-driven approach in which we synthesize images predicted to activate a given brain region using paired natural images and fMRI recordings, bypassing the need for category-specific stimuli. Our approach – Brain Diffusion for Visual Exploration ("BrainDiVE") – builds on recent generative methods by combining large-scale diffusion models with brain-guided image synthesis. Validating our method, we demonstrate the ability to synthesize preferred images with appropriate semantic specificity for well-characterized category-selective ROIs. We then show that BrainDiVE can characterize differences between ROIs selective for the same high-level category. Finally we identify novel functional subdivisions within these ROIs, validated with behavioral data. These results advance our understanding of the fine-grained functional organization of human visual cortex, and provide well-specified constraints for further examination of cortical organization using hypothesis-driven methods. Code and project site: https://www.cs.cmu.edu/~afluo/BrainDiVE

## 1 Introduction

The human visual cortex plays a fundamental role in our ability to process, interpret, and act on visual information. While previous studies have provided important evidence that regions in the higher visual cortex preferentially process complex semantic categories such as faces, places, bodies, words, and food [1, 2, 3, 4, 5, 6, 7], these important discoveries have been primarily achieved through the use of researcher-crafted stimuli. However, hand-selected, synthetic stimuli may bias the results or may not accurately capture the complexity and variability of natural scenes, sometimes leading to debates about the interpretation and validity of identified functional regions [8]. Furthermore, mapping selectivity based on responses to a fixed set of stimuli is necessarily limited, in that it can only identify selectivity for the stimulus properties that are sampled. For these reasons, data-driven methods for interpreting high-dimensional neural tuning are complementary to traditional approaches.

We introduce Brain Diffusion for Visual Exploration ("BrainDiVE"), a *generative* approach for synthesizing images that are predicted to activate a given region in the human visual cortex. Several

---

\* Co-corresponding Authors

37th Conference on Neural Information Processing Systems (NeurIPS 2023).

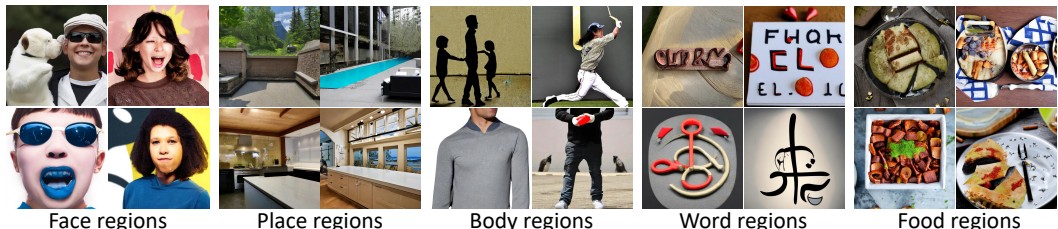

| Face regions | Place regions | Body regions | Word regions | Food regions |

Figure 1: **Images generated using BrainDiVE .** Images are generated using a diffusion model with maximization of voxels identified from functional localizer experiments as conditioning. We find that brain signals recorded via fMRI can guide the synthesis of images with high semantic specificity, strengthening the evidence for previously identified category selective regions. Select images are shown, please see below for uncurated images.

recent studies have yielded intriguing results by combining deep generative models with brain guidance [9, 10, 11]. BrainDiVE, enabled by the recent availability of large-scale fMRI datasets based on natural scene images [12, 13], allows us to further leverage state-of-the-art diffusion models in identifying fine-grained functional specialization in an objective and data-driven manner. BrainDiVE is based on image diffusion models which are typically driven by text prompts in order to generate synthetic stimuli [14]. We replace these prompts with maximization of voxels in given brain areas. The result being that the resultant synthesized images are tailored to targeted regions in higher-order visual areas. Analysis of these images enables data-driven exploration of the underlying feature preferences for different visual cortical sub-regions. Importantly, because the synthesized images are optimized to maximize the response of a given sub-region, these images emphasize and isolate critical feature preferences beyond what was present in the original stimulus images used in collecting the brain data. To validate our findings, we further performed several human behavioral studies that confirmed the semantic identities of our synthesized images.

More broadly, we establish that BrainDiVE can synthesize novel images (Figure 1) for category-selective brain regions with high semantic specificity. Importantly, we further show that Brain-DiVE can identify ROI-wise differences in selectivity that map to ecologically relevant properties. Building on this result, we are able to identify novel functional distinctions within sub-regions of existing ROIs. Such results demonstrate that BrainDiVE can be used in a data-driven manner to enable new insights into the fine-grained functional organization of the human visual cortex.

## 2    Related work

**Mapping High-Level Selectivity in the Visual Cortex.**    Certain regions within the higher visual cortex are believed to specialize in distinct aspects of visual processing, such as the perception of faces, places, bodies, food, and words [15, 3, 4, 1, 16, 17, 18, 19, 5, 20]. Many of these discoveries rely on carefully handcrafted stimuli specifically designed to activate targeted regions. However, activity under natural viewing conditions is known to be different [21]. Recent efforts using artificial neural networks as image-computable encoders/predictors of the visual pathway [22, 23, 24, 25, 26, 27, 28, 29, 30] have facilitated the use of more naturalistic stimulus sets. Our proposed method incorporates an image-computable encoding model in line with this past work.

**Deep Generative Models.**    The recent rise of learned generative models has enabled sampling from complex high dimensional distributions. Notable approaches include variational autoencoders [31, 32], generative adversarial networks [33], flows [34, 35], and score/energy/diffusion models [36, 37, 38, 39]. It is possible to condition the model on category [40, 41], text [42, 43], or images [44]. Recent diffusion models have been conditioned with brain activations to reconstruct observed images [45, 46, 47, 48, 49]. Unlike BrainDiVE, these approaches tackle reconstruction but not synthesis of novel images that are predicted to activate regions of the brain.

**Brain-Conditioned Image Generation.**    The differentiable nature of deep encoding models inspired work to create images from brain gradients in mice, macaques, and humans [50, 51, 52]. Without constraints, the images recovered are not naturalistic. Other approaches have combined deep generative models with optimization to recover natural images in macaque and humans [10, 11, 9]. Both [11, 9] utilize fMRI brain gradients combined with ImageNet trained BigGAN. In particular [11] performs end-to-end differentiable optimization by assuming a soft relaxation over the 1, 000 ImageNet classes; while [9] trains an encoder on the NSD dataset [13] and first searches for

top-classes, then performs gradient optimization within the identified classes. Both approaches are restricted to ImageNet images, which are primarily images of single objects. Our work presents major improvements by enabling the use of diffusion models [44] trained on internet-scale datasets [53] over three magnitudes larger than ImageNet. Concurrent work by [54] explore the use of gradients from macaque V4 with diffusion models, however their approach focuses on early visual cortex with grayscale image outputs, while our work focuses on higher-order visual areas and synthesize complex compositional scenes. By avoiding the search-based optimization procedures used in [9], our work is not restricted to images within a fixed class in ImageNet. Further, to the authors' knowledge we are the first work to use image synthesis methods in the identification of functional specialization in sub-parts of ROIs.

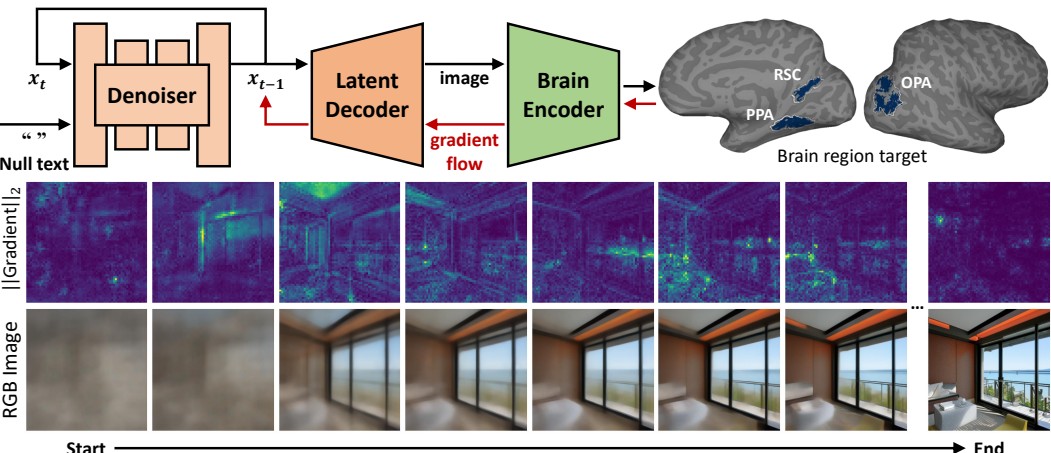

Figure 2: **Architecture of brain guided diffusion (BrainDiVE)**. **Top:** Our framework consists of two core components: **(1)** A diffusion model trained to synthesize natural images by iterative denoising; we utilize pretrained LDMs. **(2)** An encoder trained to map from images to cortical activity. Our framework can synthesize images that are predicted to activate any subset of voxels. Shown here are scene-selective regions (RSC/PPA/OPA) on the right hemisphere. **Bottom:** We visualize every 4 steps the magnitude of the gradient of the brain w.r.t. the latent and the corresponding "predicted $x_0$" [55] when targeting scene selective voxels in both hemispheres. We find clear structure emerges.

## 3 Methods

We aim to generate stimuli that maximally activate a given region in visual cortex using paired natural image stimuli and fMRI recordings. We first review relevant background information on diffusion models. We then describe how we can parameterize encoding models that map from images to brain data. Finally, we describe how our framework (Figure 2) can leverage brain signals as guidance to diffusion models to synthesize images that activate a target brain region.

### 3.1 Background on Diffusion Models

Diffusion models enable sampling from a data distribution $p(x)$ by iterative denoising. The sampling process starts with $x_T \sim \mathcal{N}(0, \mathbb{I})$, and produces progressively denoised samples $x_{T-1}, x_{T-2}, x_{T-3} \dots$ until a sample $x_0$ from the target distribution is reached. The noise level varies by timestep $t$, where the sample at each timestep is a weighted combination of $x_0$ and $\epsilon \sim \mathcal{N}(0, \mathbb{I})$, with $x_t = \sqrt{\alpha_t}x_0 + \epsilon\sqrt{1 - \alpha_t}$. The value of $\alpha$ interpolates between $\mathcal{N}(0, \mathbb{I})$ and $p(x)$.

In the noise prediction setting, an autoencoder network $\epsilon_\theta(x_t, t)$ is trained using a mean-squared error $\mathbb{E}_{(x,\epsilon,t)} \left[ \|\epsilon_\theta(x_t, t) - \epsilon\|_2^2 \right]$. In practice, we utilize a pretrained latent diffusion model (LDM) [44], with learned image encoder $E_\Phi$ and decoder $D_\Omega$, which together act as an autoencoder $\mathcal{I} \approx D_\Omega(E_\Phi(\mathcal{I}))$. The diffusion model is trained to sample $x_0$ from the latent space of $E_\Phi$.

### 3.2 Brain-Encoding Model Construction

A learned voxel-wise brain encoding model is a function $M_\theta$ that maps an image $\mathcal{I} \in \mathbb{R}^{3 \times H \times W}$ to the corresponding brain activation fMRI beta values represented as an $N$ element vector $B \in \mathbb{R}^N$: $M_\theta(\mathcal{I}) \Rightarrow B$. Past work has identified later layers in neural networks as the best predictors of higher visual cortex [30, 56], with CLIP trained networks among the highest performing brain

encoders [28, 57]. As our target is the higher visual cortex, we utilize a two component design for our encoder. The first component consists of a CLIP trained image encoder which outputs a $K$ dimensional vector as the latent embedding. The second component is a linear adaptation layer $W \in \mathcal{R}^{N \times K}, b \in \mathcal{R}^N$, which maps euclidean normalized image embeddings to brain activation.

$$B \approx M_\theta(\mathcal{I}) = W \times \frac{\texttt{CLIP}_{\texttt{img}}(\mathcal{I})}{\|\texttt{CLIP}_{\texttt{img}}(\mathcal{I})\|_2} + b$$

Optimal $W^*, b^*$ are found by optimizing the mean squared error loss over images. We observe that use of a normalized CLIP embedding improves stability of gradient magnitudes w.r.t. the image.

### 3.3 Brain-Guided Diffusion Model

BrainDiVE seeks to generate images conditioned on maximizing brain activation in a given region. In conventional text-conditioned diffusion models, the conditioning is done in one of two ways. The first approach modifies the function $\epsilon_\theta$ to further accept a conditioning vector $c$, resulting in $\epsilon_\theta(x_t, t, c)$. The second approach uses a contrastive trained image-to-concept encoder, and seeks to maximize a similarity measure with a text-to-concept encoder.

Conditioning on activation of a brain region using the first approach presents difficulties. We do not know *a priori* the distribution of other non-targeted regions in the brain when a target region is maximized. Overcoming this problem requires us to either have a prior $p(B)$ that captures the joint distribution for all voxels in the brain, to ignore the joint distribution that can result in catastrophic effects, or to use a handcrafted prior that may be incorrect [47]. Instead, we propose to condition the diffusion model via our image-to-brain encoder. During inference we perturb the denoising process using the gradient of the brain encoder *maximization* objective, where $\gamma$ is a scale, and $S \subseteq N$ are the set of voxels used for guidance. We seek to maximize the average activation of $S$ predicted by $M_\theta$:

$$\epsilon'_{theta} = \epsilon_{theta} - \sqrt{1 - \alpha_t}\nabla_{x_t}\left(\frac{\gamma}{|S|}\sum_{i \in S} M_\theta(D_\Omega(x'_t))_i\right)$$

Like [14, 58, 59], we observe that convergence using the current denoised $x_t$ is poor without changes to the guidance. This is because the current image (latent) is high noise and may lie outside of the natural image distribution. We instead use a weighted reformulation with an euler approximation [55, 59] of the final image:

$$\hat{x}_0 = \frac{1}{\sqrt{\alpha}}(x_t - \sqrt{1 - \alpha}\epsilon_t)$$
$$x'_t = (\sqrt{1 - \alpha})\hat{x}_0 + (1 - \sqrt{1 - \alpha})x_t$$

By combining an image diffusion model with a differentiable encoding model of the brain, we are able to generate images that seek to maximize activation for any given brain region.

## 4 Results

In this section, we use BrainDiVE to highlight the semantic selectivity of pre-identified category-selective voxels. We then show that our model can capture subtle differences in response properties between ROIs belonging to the same broad category-selective network. Finally, we utilize Brain-DiVE to target finer-grained sub-regions within existing ROIs, and show consistent divisions based on semantic and visual properties. We quantify these differences in selectivity across regions using human perceptual studies, which confirm that BrainDiVE images can highlight differences in tuning properties. These results demonstrate how BrainDiVE can elucidate the functional properties of human cortical populations, making it a promising tool for exploratory neuroscience.

### 4.1 Setup

We utilize the Natural Scenes Dataset (NSD; [13]), which consists of whole-brain 7T fMRI data from 8 human subjects, 4 of whom viewed $10,000$ natural scene images repeated $3\times$. These subjects, S1, S2, S5, and S7, are used for analyses in the main paper (see Supplemental for results for additional subjects). All images are from the MS COCO dataset. We use beta-weights (activations) computed using GLMSingle [60] and further normalize each voxel to $\mu = 0, \sigma = 1$ on a per-session basis. We average the fMRI activation across repeats of the same image within a subject. The $\sim 9,000$ unique images for each subject ([13]) are used to train the brain encoder for each subject, with the remaining $\sim 1,000$ shared images used to evaluate $R^2$. Image generation is on a per-subject basis and done

on an `Nvidia V100` using $1,500$ compute hours. As the original category ROIs in NSD are very generous, we utilize a stricter $t > 2$ threshold to reduce overlap unless otherwise noted. The final category and ROI masks used in our experiments are derived from the logical `AND` of the official NSD masks with the masks derived from the official $t$-statistics.

We utilize `stable-diffusion-2-1-base`, which produces images of $512 \times 512$ resolution using $\epsilon$-prediction. Following best practices, we use multi-step 2nd order DPM-Solver++ [61] with 50 steps and apply $0.75$ SAG [62]. We set step size hyperparameter $\gamma = 130.0$. Images are resized to $224 \times 224$ for the brain encoder. "" (null prompt) is used as the input prompt, thus the diffusion performs unconditional generation without brain guidance. For the brain encoder we use `ViT-B/16`, for CLIP probes we use `CoCa ViT-L/14`. These are the highest performing `LAION-2B` models of a given size provided by `OpenCLIP` [63, 64, 65, 66]. We train our brain encoders on each human subject separately to predict the activation of all higher visual cortex voxels. See Supplemental for visualization of test time brain encoder $R^2$. To compare images from different ROIs and sub-regions (OFA/FFA in 4.3, two clusters in 4.4), we asked human evaluators select which of two image groups scored higher on various attributes. We used 100 images from each group randomly split into 10 non-overlapping subgroups. Each human evaluator performed 80 comparisons, across 10 splits, 4 NSD subjects, and for both fMRI and generated images. See Supplemental for standard error of responses. Human evaluators provided written informed consent and were compensated at \$12.00/hour. The study protocol was approved by the institutional review board at the authors' institution.

## 4.2 Broad Category-Selective Networks

In this experiment, we target large groups of category-selective voxels which can encompass more than one ROI (Figure 3). These regions have been previously identified as selective for broad semantic categories, and this experiment validates our method using these identified regions. The face-, place-, body-, and word- selective ROIs are identified with standard localizer stimuli [67]. The food-selective voxels were obtained from [5].

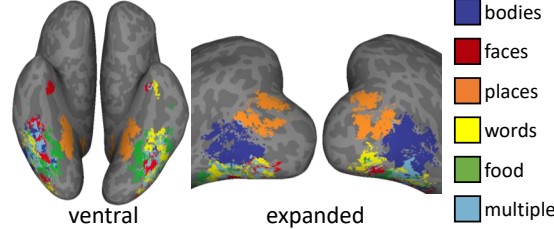

Figure 3: **Visualizing category-selective voxels in S1.** See text for details on how category selectivity was defined.

The same voxels were used to select the top activating NSD images (referred to as "NSD") and to guide the generation of BrainDiVE images.

In Figures 4 we visualize, for place-, face-, word-, and body- selective voxels, the top-5 out of $10,000$ images from the fMRI stimulus set (NSD), and the top-5 images out of $1,000$ total images as evaluated by the encoding component of BrainDiVE. For food selective voxels, the top-10 are visualized. A visual inspection indicates that our method is able to generate diverse images that semantically represent the target category. We further use CLIP to perform semantic probing of the images, and force the images to be classified into one of five categories. We measure the percentage of images that match the preferred category for a given set of voxels (Table 1). We find that our top-10% and 20% of images exceed the top-1% and 2% of natural images in accuracy, indicating our method has high semantic specificity.

| | Faces | | Places | | Bodies | | Words | | Food | | Mean | |
|---|---|---|---|---|---|---|---|---|---|---|---|---|
| | S1↑ | S2↑ | S1↑ | S2↑ | S1↑ | S2↑ | S1↑ | S2↑ | S1↑ | S2↑ | S1↑ | S2↑ |
| NSD all stim | 17.4 | 17.2 | 29.9 | 29.5 | 31.6 | 31.8 | 10.3 | 10.6 | 10.8 | 10.9 | 20.0 | 20.0 |
| NSD top-200 | 42.5 | 41.5 | 66.5 | 80.0 | 56.0 | 65.0 | 31.5 | 34.5 | 68.0 | 85.5 | 52.9 | 61.3 |
| NSD top-100 | 40.0 | 45.0 | 68.0 | 79.0 | 49.0 | 60.0 | 30.0 | 49.0 | 78.0 | 85.0 | 53.0 | 63.6 |
| BrainDiVE-200 | **69.5** | **70.0** | **97.5** | **100** | **75.5** | 68.5 | **60.0** | 57.5 | 89.0 | 94.0 | **78.3** | 75.8 |
| BrainDiVE-100 | 61.0 | 68.0 | 97.0 | **100** | 75.0 | **69.0** | **60.0** | 62.0 | 92.0 | 95.0 | 77.0 | **78.8** |

Table 1: **Evaluating semantic specificity with zero-shot CLIP classification.** We use CLIP to classify images from each ROI into five semantic categories: face/place/body/word/food. Shown is the percentage where the classified category of the image matches the preferred category of the brain region. We show this for each subject's entire NSD stimulus set ($10,000$ images for S1&S2); the top-200 and top-100 images (top-2% and top-1%) evaluated by mean true fMRI beta, and the top-200 and top-100 (20% and 10%) of BrainDiVE images as self-evaluated by the encoding component of BrainDiVE. BrainDiVE generates images with higher semantic specificity than the top 1% of natural images for each brain region.

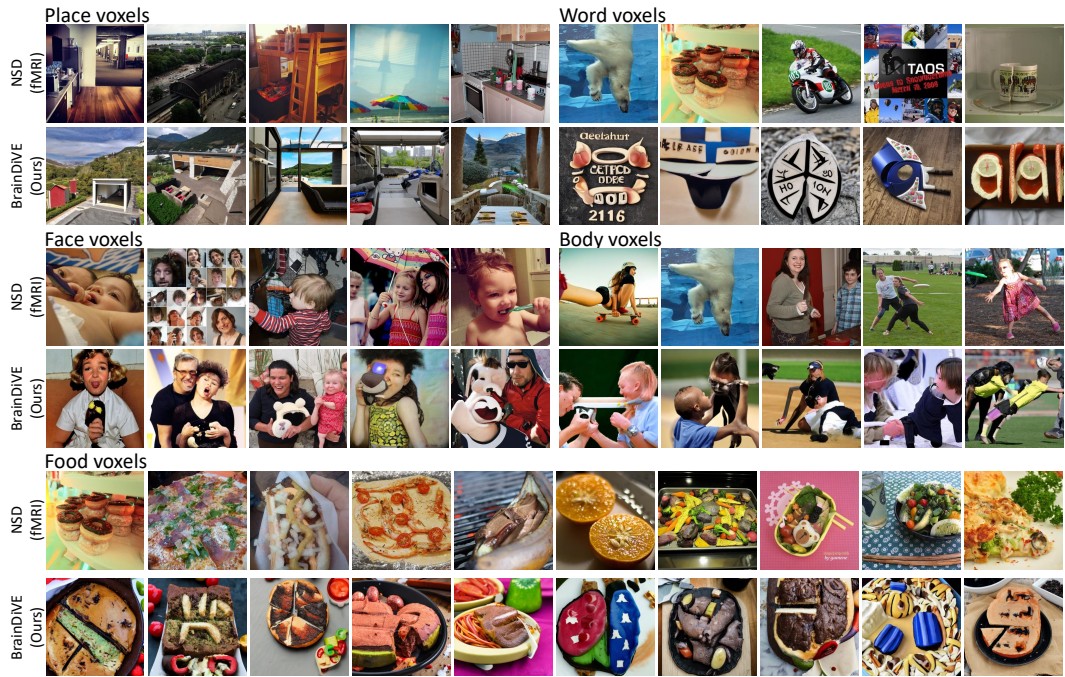

Figure 4: **Results for category selective voxels (S1).** We identify the top-5 images from the stimulus set or generated by our method with highest average activation in each set of category selective voxels for the face/place/word/body categories, and the top-10 images for the food selective voxels.

| Which ROI has more... | photorealistic faces | | | | animals | | | | abstract shapes/lines | | | |
|---|---|---|---|---|---|---|---|---|---|---|---|---|
| | S1 | S2 | S5 | S7 | S1 | S2 | S5 | S7 | S1 | S2 | S5 | S7 |
| FFA-NSD | **45** | **43** | **34** | **41** | 34 | 34 | 17 | 15 | 21 | 6 | 14 | 22 |
| OFA-NSD | 25 | 22 | 21 | 18 | **47** | **36** | **65** | **65** | 24 | 44 | 28 | 25 |
| FFA-BrainDiVE | **79** | **89** | **60** | **52** | 17 | 13 | 21 | 19 | 6 | 11 | 18 | 20 |
| OFA-BrainDiVE | 11 | 4 | 15 | 22 | **71** | **61** | **52** | **50** | 80 | 79 | 40 | 39 |

Table 2: **Human evaluation of the difference between face-selective ROIs.** Evaluators compare groups of images corresponding to OFA and FFA; comparisons are done within GT and generated images respectively. Questions are posed as: "Which group of images has more X?"; options are FFA/OFA/Same. Results are in %. Note that the "Same" responses are not shown; responses across all three options sum to 100.

## 4.3 Individual ROIs

In this section, we apply our method to individual ROIs that are selective for the same broad semantic category. We focus on the occipital face area (OFA) and fusiform face area (FFA), as initial tests suggested little differentiation between ROIs within the place-, word-, and body- selective networks. In this experiment, we also compare our results against the top images for FFA and OFA from NeuroGen [9], using the top 100 out of 500 images provided by the authors. Following NeuroGen, we also generate 500 total images, targeting FFA and OFA separately (Figure 5). We observe that both diffusion-generated and NSD images have very high face content in FFA, whereas NeuroGen has higher animal face content. In OFA, we observe both NSD and BrainDiVE images have a strong face component, although we also observe text selectivity in S2 and animal face selectivity in S5. Again NeuroGen predicts a higher animal component than face for S5. By avoiding the use of fixed categories, BrainDiVE images are more diverse than those of NeuroGen. This trend of face and animals appears at $t > 2$ and the much stricter $t > 5$ threshold for identifying face-selective voxels ($t > 5$ used for visualization/evaluation). The differences in images synthesized by BrainDiVE for FFA and OFA are consistent with past work suggesting that FFA represents faces at a higher level of abstraction than OFA, while OFA shows greater selectivity to low-level face features and sub-components, which could explain its activation by off-target categories [68, 69, 70].

To quantify these results, we perform a human study where subjects are asked to compare the top-100 images between FFA & OFA, for both NSD and generated images. Results are shown in Table 2.

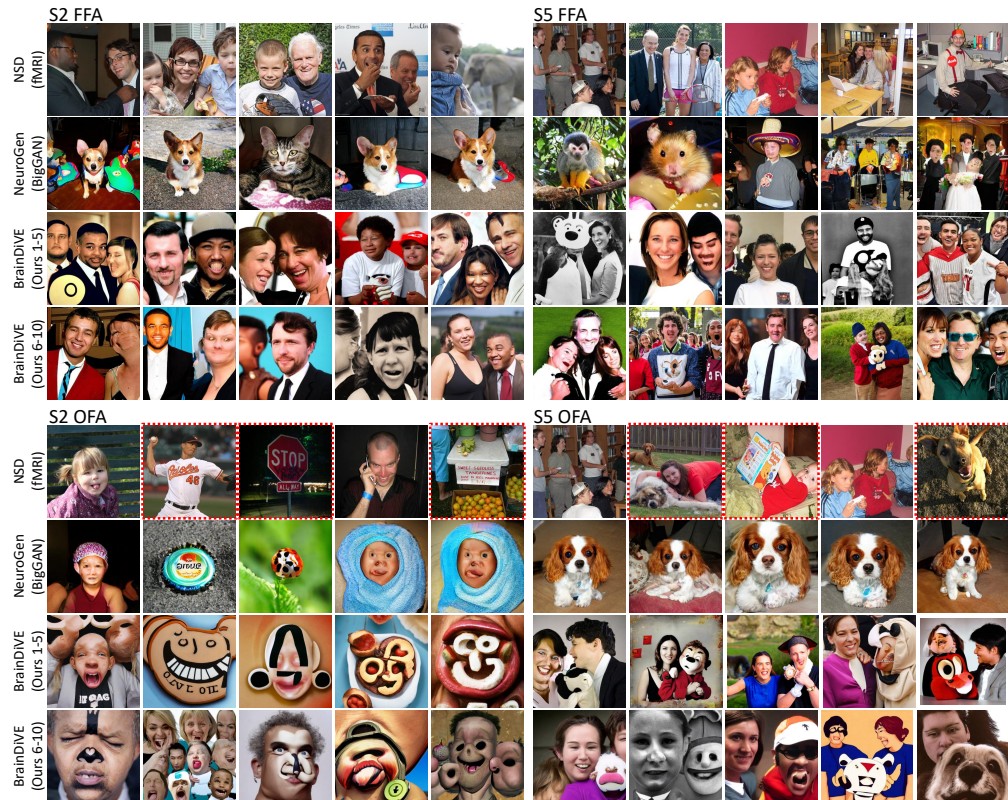

Figure 5: **Results for face-selective ROIs.** For each ROI (OFA, FFA) we visualize the top-5 images from NSD and NeuroGen, and the top-10 from BrainDiVE. NSD images are selected using the fMRI betas averaged within each ROI. NeuroGen images are ranked according to their official predicted ROI activity means. BrainDiVE images are ranked using our predicted ROI activities from 500 images. Red outlines in the NSD images indicate examples of responsiveness to non-face content.

We find that OFA consistently has higher animal and abstract content than FFA. Most notably, this difference is on average more pronounced in the images from BrainDiVE, indicating that our approach is able to highlight subtle differences in semantic selectivity across regions.

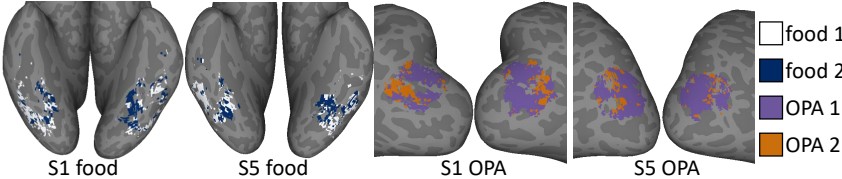

Figure 6: **Clustering within the food ROI and within OPA**. Clustering of encoder model weights for each region is shown for two example subjects on an inflated cortical surface.

## 4.4 Semantic Divisions within ROIs

In this experiment, we investigate if our model can identify novel sub-divisions within existing ROIs. We first perform clustering on normalized per-voxel encoder weights using vmf-clustering [71]. We find consistent cosine difference between the cluster centers in the food-selective ROI as well as in the occipital place area (OPA), clusters shown in Figure 6. In all four subjects, we observe a relatively consistent anterior-posterior split of OPA. While the clusters within the food ROI vary more anatomically, each subject appears to have a more medial and a more lateral cluster. We visualize the images for the two food clusters in Figure 7, and for the two OPA clusters in Figure 8. We observe that for both the food ROI and OPA, the BrainDiVE-generated images from each cluster have noticeable differences in their visual and semantic properties. In particular, the BrainDiVE images from food cluster-2 have much higher color saturation than those from cluster-1, and also have more objects

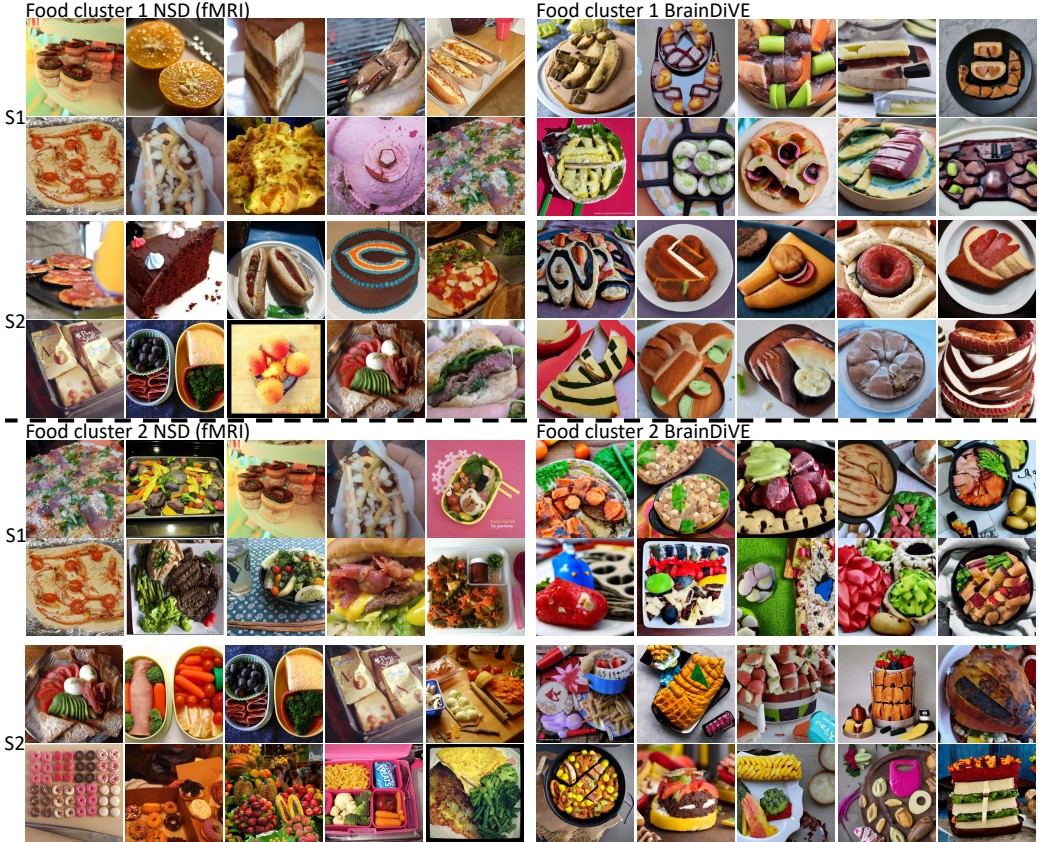

Figure 7: **Comparing results across the food clusters.** We visualize top-10 NSD fMRI (out of 10,000) and diffusion images (out of 500) for *each cluster*. While the first cluster largely consists of processed foods, the second cluster has more visible high color saturation foods, and more vegetables/fruit like objects. BrainDiVE helps highlight the differences between clusters.

| Which cluster is more ... | vegetables/fruits | | | | healthy | | | | colorful | | | | far away | | | |
|---|---|---|---|---|---|---|---|---|---|---|---|---|---|---|---|---|
| | S1 | S2 | S5 | S7 | S1 | S2 | S5 | S7 | S1 | S2 | S5 | S7 | S1 | S2 | S5 | S7 |
| Food-1 NSD | 17 | 21 | 27 | 36 | 28 | 22 | 29 | 40 | 19 | 18 | 13 | 27 | 32 | 24 | 23 | 28 |
| Food-2 NSD | **65** | **56** | **56** | **49** | **50** | **47** | **54** | **45** | **42** | **52** | **53** | **42** | **34** | **39** | **36** | **42** |
| Food-1 BrainDiVE | 11 | 10 | 8 | 11 | 15 | 16 | 20 | 17 | 6 | 9 | 11 | 16 | 24 | 18 | 27 | 18 |
| Food-2 BrainDiVE | **80** | **75** | **67** | **64** | **68** | **68** | **46** | **51** | **79** | **82** | **65** | **61** | **39** | **51** | **39** | **40** |

Table 3: **Human evaluation of the difference between food clusters**. Evaluators compare groups of images corresponding to food cluster 1 (Food-1) and food cluster 2 (Food-2), with questions posed as "Which group of images has/is more X?". Comparisons are done within NSD and generated images respectively. Note that the "Same" responses are not shown; responses across all three options sum to 100. Results are in %.

that resemble fruits and vegetables. In contrast, food cluster-1 generally lacks vegetables and mostly consist of bread-like foods. In OPA, cluster-1 is dominated by indoor scenes (rooms, hallways), while 2 is overwhelmingly outdoor scenes, with a mixture of natural and man-made structures viewed from a far perspective. Some of these differences are also present in the NSD images, but the differences appear to be highlighted in the generated images.

To confirm these effects, we perform a human study (Table 3, Table 4) comparing the images from different clusters in each ROI, for both NSD and generated images. As expected from visual inspection of the images, we find that food cluster-2 is evaluated to have higher vegetable/fruit content, judged to be healthier, more colorful, and slightly more distant than food cluster-1. We find that OPA cluster-1 is evaluated to be more angular/geometric, include more indoor scenes, to be less natural and consisting of less distant scenes. Again, while these trends are present in the NSD images, they are more pronounced with the BrainDiVE images. This not only suggests that our method has uncovered differences in semantic selectivity within pre-existing ROIs, but also reinforces the ability of BrainDiVE to identify and highlight core functional differences across visual cortex regions.

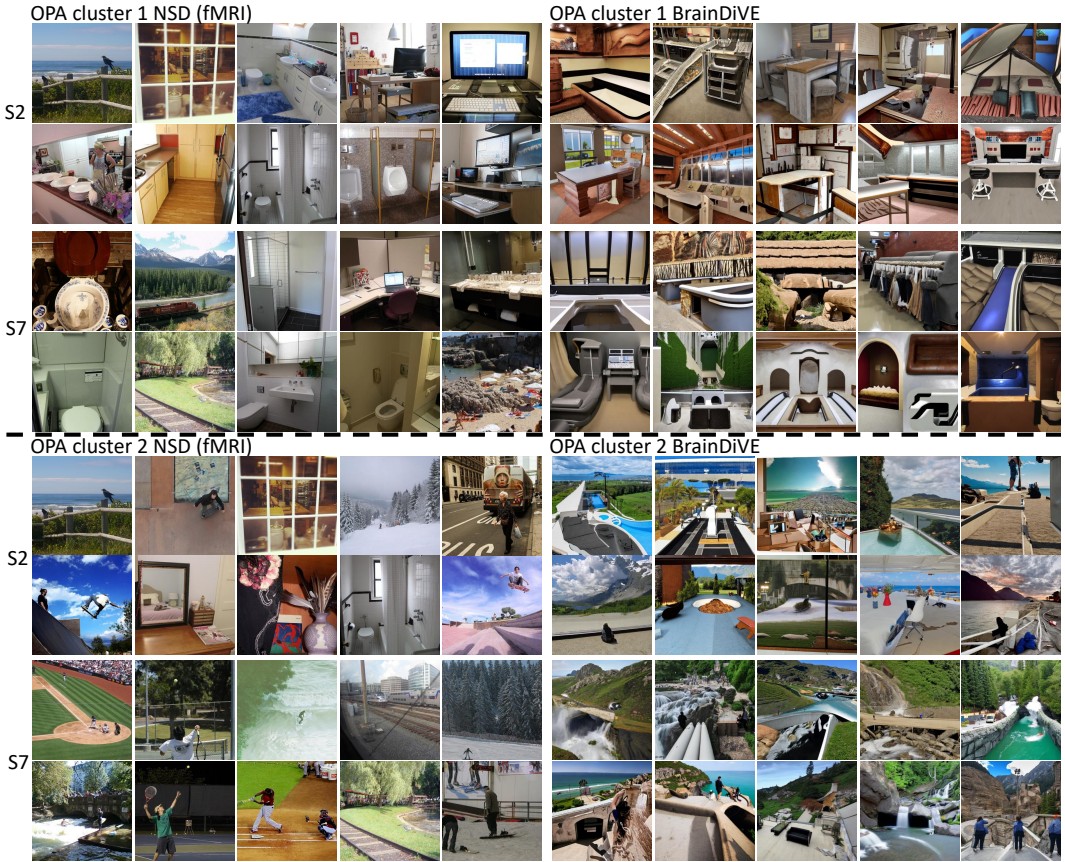

Figure 8: **Comparing results across the OPA clusters.** We visualize top-10 NSD fMRI (out of 10,000) and diffusion images (out of 500) for *each cluster*. While both consist of scene images, the first cluster have more indoor scenes, while the second has more outdoor scenes. The BrainDiVE images help highlight the differences in semantic properties.

| Which cluster is more... | angular/geometric | | | | indoor | | | | natural | | | | far away | | | |
|---|---|---|---|---|---|---|---|---|---|---|---|---|---|---|---|---|
| | S1 | S2 | S5 | S7 | S1 | S2 | S5 | S7 | S1 | S2 | S5 | S7 | S1 | S2 | S5 | S7 |
| OPA-1 NSD | **45** | **58** | **49** | **51** | **71** | **88** | **80** | **79** | 14 | 3 | 9 | 10 | 10 | 1 | 6 | 8 |
| OPA-2 NSD | 13 | 12 | 14 | 16 | 7 | 8 | 11 | 14 | **73** | **89** | **71** | **81** | **69** | **93** | **81** | **85** |
| OPA-1 BrainDiVE | **76** | **87** | **88** | **76** | **89** | **90** | **90** | **85** | 6 | 6 | 9 | 6 | 1 | 3 | 3 | 8 |
| OPA-2 BrainDiVE | 12 | 3 | 4 | 10 | 7 | 7 | 5 | 8 | **91** | **91** | **83** | **90** | **97** | **92** | **91** | **88** |

Table 4: **Human evaluation of the difference between OPA clusters**. Evaluators compare groups of images corresponding to OPA cluster 1 (OPA-1) and OPA cluster 2 (OPA-2), with questions posed as "Which group of images is more X?". Comparisons are done within NSD and generated images respectively. Note that the "Same" responses are not shown; responses across all three options sum to 100. Results are in %.

## 5 Discussion

**Limitations and Future Work**  Here, we show that BrainDiVE generates diverse and realistic images that can probe the human visual pathway. This approach relies on existing large datasets of natural images paired with brain recordings. In that the evaluation of synthesized images is necessarily qualitative, it will be important to validate whether our generated images and candidate features derived from these images indeed maximize responses in their respective brain areas. As such, future work should involve the collection of human fMRI recordings using both our synthesized images and more focused stimuli designed to test our qualitative observations. Future work may also explore the images generated when BrainDiVE is applied to additional sub-region, new ROIs, or mixtures of ROIs.

**Conclusion**  We introduce a novel method for guiding diffusion models using brain activations – BrainDiVE – enabling us to leverage generative models trained on internet-scale image datasets for

data driven explorations of the brain. This allows us to better characterize fine-grained preferences across the visual system. We demonstrate that BrainDiVE can accurately capture the semantic selectivity of existing characterized regions. We further show that BrainDiVE can capture subtle differences between ROIs within the face selective network. Finally, we identify and highlight fine-grained subdivisions within existing food and place ROIs, differing in their selectivity for mid-level image features and semantic scene content. We validate our conclusions with extensive human evaluation of the images.

## 6   Acknowledgements

This work used Bridges-2 at Pittsburgh Supercomputing Center through allocation SOC220017 from the Advanced Cyberinfrastructure Coordination Ecosystem: Services & Support (ACCESS) program, which is supported by National Science Foundation grants #2138259, #2138286, #2138307, #2137603, and #2138296. We also thank the Carnegie Mellon University Neuroscience Institute for support.

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

# Supplementary Material:
# Brain Diffusion for Visual Exploration

## 7 Broader impacts

Our work introduces a method where brain responses - as measured by fMRI - can be used to guide diffusion models for image synthesis (BrainDiVE). We applied BrainDiVE to probe the representation of high-level semantic information in the human visual cortex. BrainDiVE relies on pretrained `stable-diffusion-2-1` and will necessarily reflect the biases in the data used to train these models. However, given the size and diversity of this training data, BrainDiVE may reveal data-driven principles of cortical organization that are unlikely to have been identified using more constrained, hypothesis-driven experiments. As such, our work advances our current understanding of the human visual cortex and, with larger and more sensitive neuroscience datasets, may be utilized to facilitate future fine-grained discoveries regarding neural coding, which can then be validated using hypothesis-driven experiments.

# 8 Visualization of each subject's category selective voxel images

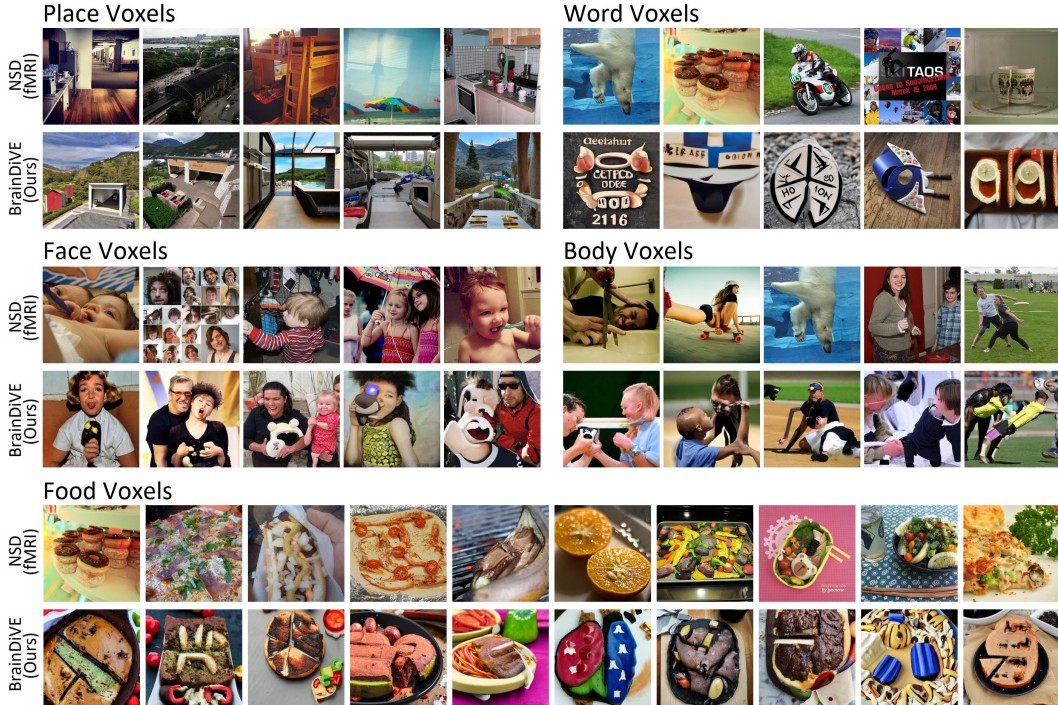

Figure S.1: **Results for category selective voxels (S1).** We identify the top-5 images from the stimulus set or generated by our method with highest average activation in each set of category selective voxels for the face/place/word/body categories, and the top-10 images for the food selective voxels. Note the top NSD body voxel image for S1 was omitted from the main paper due to content.

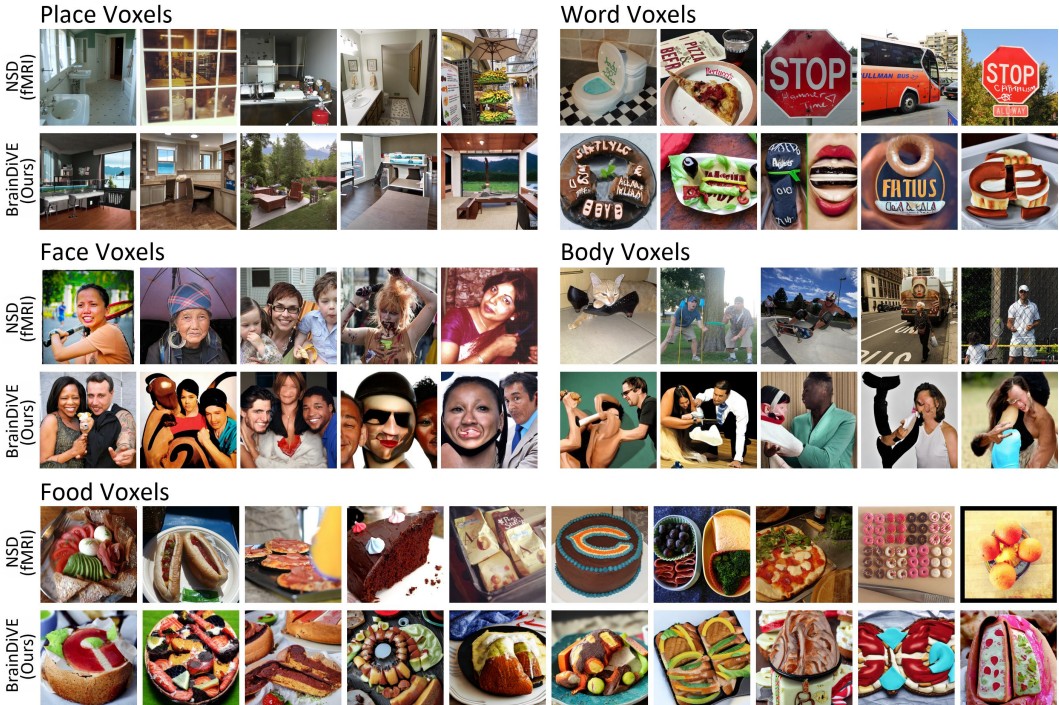

Figure S.2: **Results for category selective voxels (S2).** We identify the top-5 images from the stimulus set or generated by our method with highest average activation in each set of category selective voxels for the face/place/word/body categories, and the top-10 images for the food selective voxels.

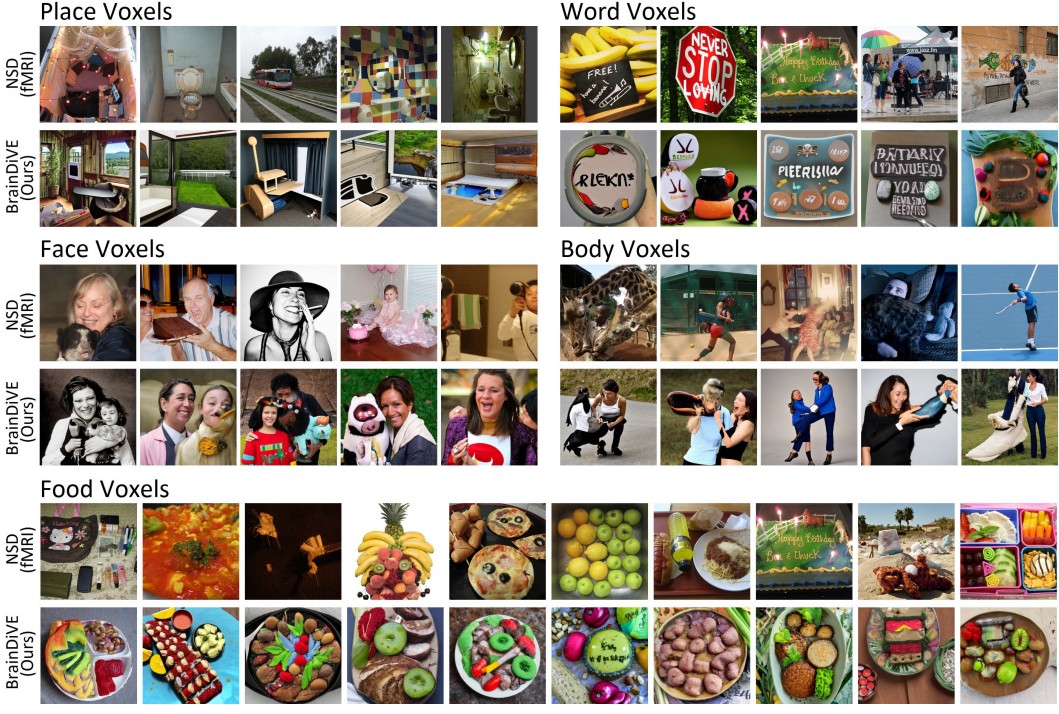

Figure S.3: **Results for category selective voxels (S3).** We identify the top-5 images from the stimulus set or generated by our method with highest average activation in each set of category selective voxels for the face/place/word/body categories, and the top-10 images for the food selective voxels.

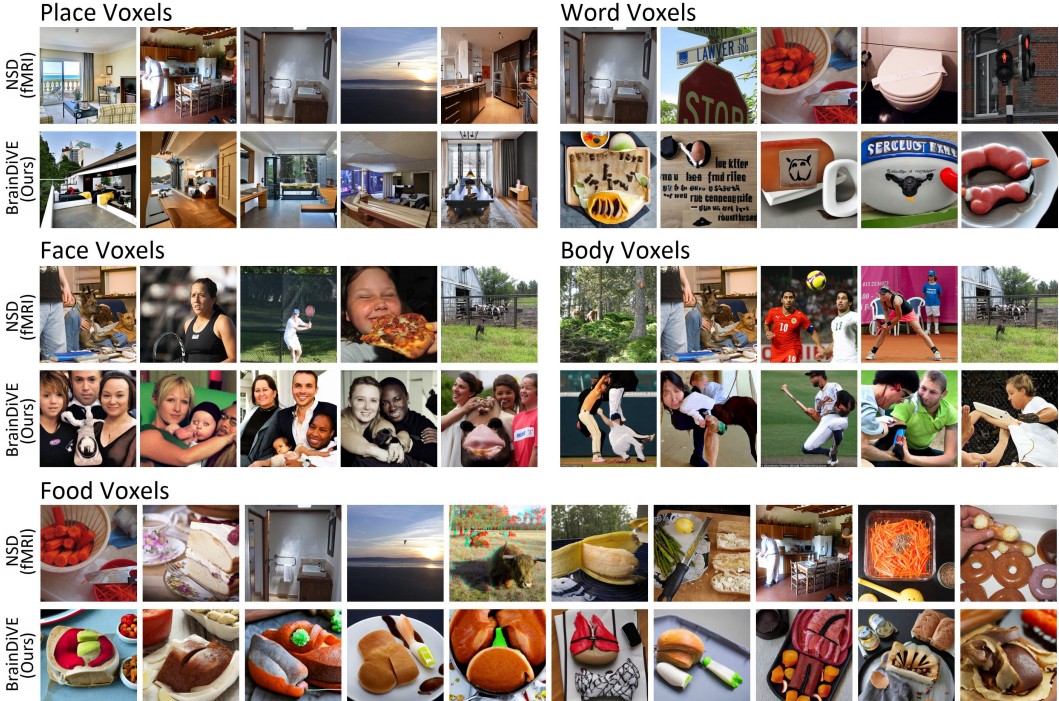

Figure S.4: **Results for category selective voxels (S4).** We identify the top-5 images from the stimulus set or generated by our method with highest average activation in each set of category selective voxels for the face/place/word/body categories, and the top-10 images for the food selective voxels.

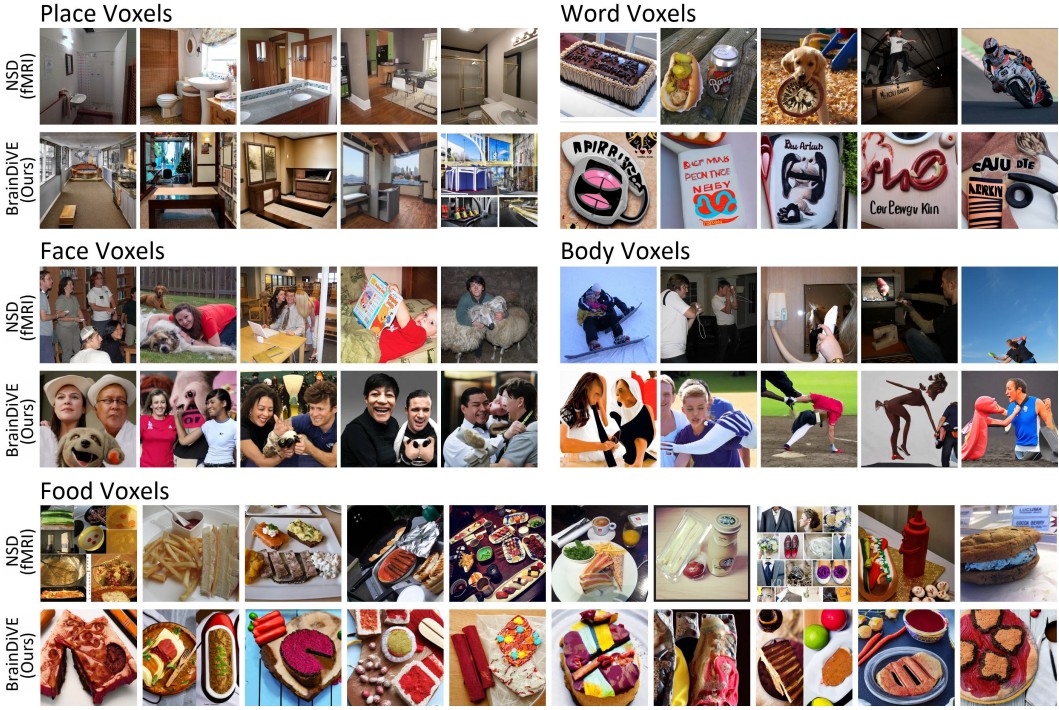

Figure S.5: **Results for category selective voxels (S5).** We identify the top-5 images from the stimulus set or generated by our method with highest average activation in each set of category selective voxels for the face/place/word/body categories, and the top-10 images for the food selective voxels.

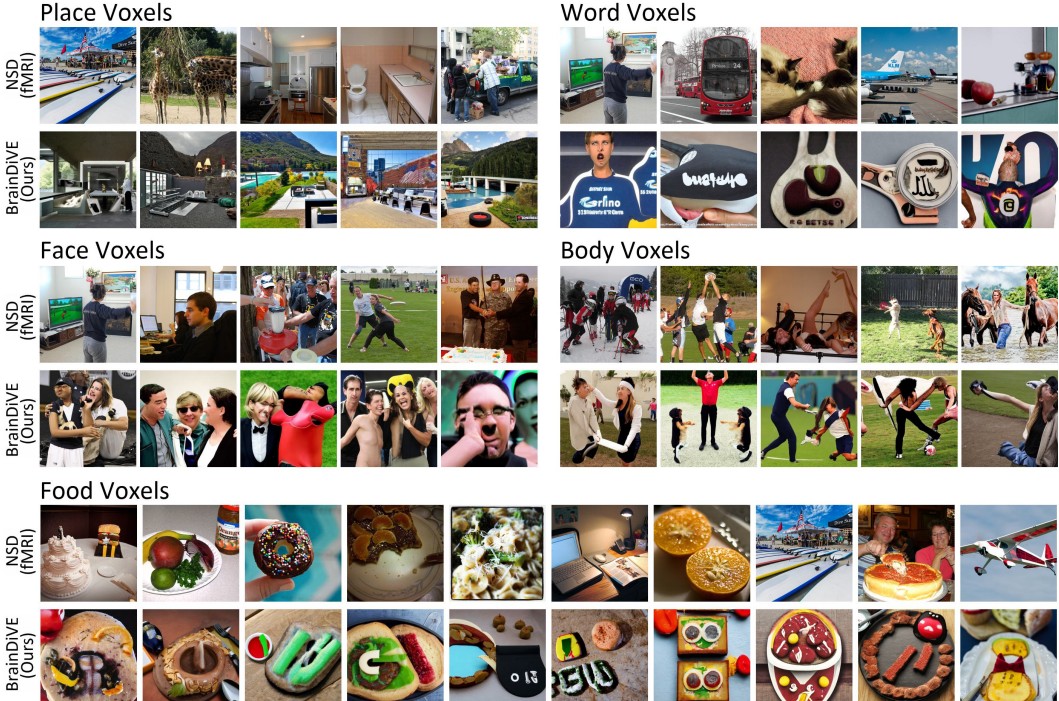

Figure S.6: **Results for category selective voxels (S6).** We identify the top-5 images from the stimulus set or generated by our method with highest average activation in each set of category selective voxels for the face/place/word/body categories, and the top-10 images for the food selective voxels.

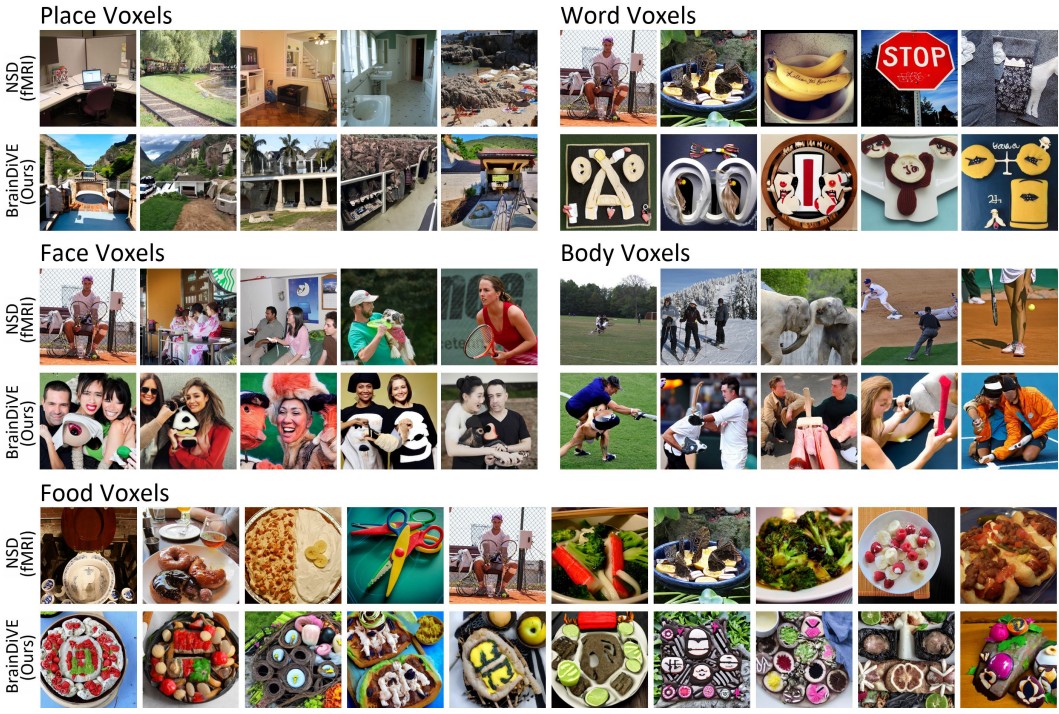

Figure S.7: **Results for category selective voxels (S7).** We identify the top-5 images from the stimulus set or generated by our method with highest average activation in each set of category selective voxels for the face/place/word/body categories, and the top-10 images for the food selective voxels.

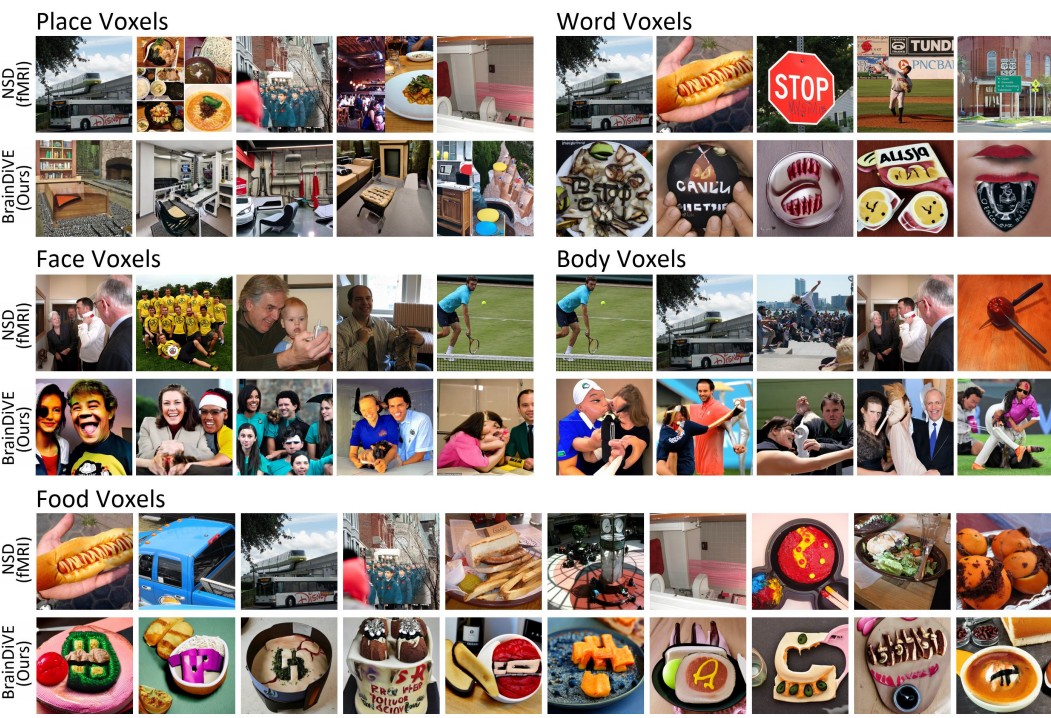

Figure S.8: **Results for category selective voxels (S8).** We identify the top-5 images from the stimulus set or generated by our method with highest average activation in each set of category selective voxels for the face/place/word/body categories, and the top-10 images for the food selective voxels.

# 9 CLIP zero-shot classification

In this section we show the CLIP classification results for S1 – S8, where Table S.1 in this Supplementary material matches that of Table 1 in the main paper. We use CLIP [63] to classify images from each ROI into five semantic categories: face/place/body/word/food. Shown is the percentage where the classified category of the image matches the preferred category of the brain region. We show this for the top-200 and top-100 images (top-2% and top-1%) evaluated by mean true fMRI beta, and the top-200 and top-100 (20% and 10%) of BrainDiVE images as self-evaluated by the encoding component of BrainDiVE. Please see Supplementary Section 15 for the prompts we use for CLIP classification.

|  | Faces | | Places | | Bodies | | Words | | Food | | Mean | |
|---|---|---|---|---|---|---|---|---|---|---|---|---|
|  | S1↑ | S2↑ | S1↑ | S2↑ | S1↑ | S2↑ | S1↑ | S2↑ | S1↑ | S2↑ | S1↑ | S2↑ |
| NSD top-200 | 42.5 | 41.5 | 66.5 | 80.0 | 56.0 | 65.0 | 31.5 | 34.5 | 68.0 | 85.5 | 52.9 | 61.3 |
| NSD top-100 | 40.0 | 45.0 | 68.0 | 79.0 | 49.0 | 60.0 | 30.0 | 49.0 | 78.0 | 85.0 | 53.0 | 63.6 |
| BrainDiVE-200 | **69.5** | **70.0** | **97.5** | **100** | **75.5** | 68.5 | **60.0** | 57.5 | 89.0 | 94.0 | **78.3** | 75.8 |
| BrainDiVE-100 | 61.0 | 68.0 | **97.0** | **100** | 75.0 | **69.0** | **60.0** | 62.0 | 92.0 | 95.0 | 77.0 | **78.8** |

Table S.1: **Evaluating semantic specificity with zero-shot CLIP classification for S1 and S2**

|  | Faces | | Places | | Bodies | | Words | | Food | | Mean | |
|---|---|---|---|---|---|---|---|---|---|---|---|---|
|  | S3↑ | S4↑ | S3↑ | S4↑ | S3↑ | S4↑ | S3↑ | S4↑ | S3↑ | S4↑ | S3↑ | S4↑ |
| NSD top-200 | 33.0 | 39.0 | 74.5 | 71.5 | 57.9 | 47.5 | 27.0 | 20.5 | 49.5 | 53.5 | 48.4 | 46.4 |
| NSD top-100 | 38.0 | 41.0 | 81.0 | 72.0 | 60.0 | 49.0 | 30.0 | 25.0 | 46.0 | 57.9 | 51.0 | 49.0 |
| BrainDiVE-200 | **67.5** | **73.5** | 99.0 | **100** | **59.0** | 66.5 | **61.0** | 31.0 | 85.0 | 89.0 | 74.3 | 72.0 |
| BrainDiVE-100 | 67.0 | 71.0 | **100** | **100** | **59.0** | **72.0** | **61.0** | **34.0** | **89.0** | **93.0** | **75.2** | **74.0** |

Table S.2: **Evaluating semantic specificity with zero-shot CLIP classification for S3 and S4**

|  | Faces | | Places | | Bodies | | Words | | Food | | Mean | |
|---|---|---|---|---|---|---|---|---|---|---|---|---|
|  | S5↑ | S6↑ | S5↑ | S6↑ | S5↑ | S6↑ | S5↑ | S6↑ | S5↑ | S6↑ | S5↑ | S6↑ |
| NSD top-200 | 41.0 | 38.5 | 89.5 | 56.9 | 57.9 | 56.5 | 33.5 | 34.0 | 77.0 | 55.5 | 59.8 | 48.3 |
| NSD top-100 | 45.0 | 46.0 | 93.0 | 55.0 | 54.0 | 61.0 | 33.0 | 32.0 | 85.0 | 56.9 | 62.0 | 50.2 |
| BrainDiVE-200 | **67.0** | **63.0** | 99.5 | 96.0 | 74.0 | 66.0 | 75.0 | 68.0 | 83.5 | 79.0 | 79.8 | 74.4 |
| BrainDiVE-100 | 64.0 | 57.9 | **100** | **99.0** | **77.0** | **72.0** | **80.0** | **75.0** | **87.0** | **83.0** | **81.6** | **77.4** |

Table S.3: **Evaluating semantic specificity with zero-shot CLIP classification for S5 and S6**

|  | Faces | | Places | | Bodies | | Words | | Food | | Mean | |
|---|---|---|---|---|---|---|---|---|---|---|---|---|
|  | S7↑ | S8↑ | S7↑ | S8↑ | S7↑ | S8↑ | S7↑ | S8↑ | S7↑ | S8↑ | S7↑ | S8↑ |
| NSD top-200 | 38.5 | 34.0 | 71.0 | 57.5 | 61.0 | 56.5 | 20.5 | 24.5 | 52.0 | 36.5 | 48.6 | 41.8 |
| NSD top-100 | 35.0 | 36.0 | 76.0 | 48.0 | 63.0 | 61.0 | 26.0 | 21.0 | 56.0 | 37.0 | 51.2 | 40.6 |
| BrainDiVE-200 | **73.0** | **77.5** | 93.5 | **94.5** | 65.0 | 64.5 | **31.0** | 56.5 | 85.5 | 55.5 | **69.6** | 69.7 |
| BrainDiVE-100 | 69.0 | 72.0 | **94.0** | 94.0 | **65.0** | **67.0** | 25.0 | 56.0 | **92.0** | **74.0** | 69.0 | **72.6** |

Table S.4: **Evaluating semantic specificity with zero-shot CLIP classification for S7 and S8.**

# 10 Image gradients and synthesis process

In this section, we show examples of the image at each step of the synthesis process. We perform this visualization for face-, place-, body-, word-, and food- selective voxels. Two visualizations are shown for each set of voxels, we use S1 for all visualizations in this section. The diffusion model is guided only by the objective of maximizing a given set of voxels. We observe that coarse image structure emerges very early on from brain guidance. Furthermore, the gradient and diffusion model sometimes work against each other. For example in Figure S.14 for body voxels, the brain gradient induces the addition of an extra arm, while the diffusion has already generated three natural bodies. Or in Figure S.15 for word voxels, where the brain gradient attempts to add horizontal words, but they are warped by the diffusion model. Future work could explore early guidance only, as described in "SDEdit" and "MagicMix" [72, 73].

## 10.1 Face voxels

We show examples where the end result contains multiple faces (Figure S.9), or a single face (Figure S.10).

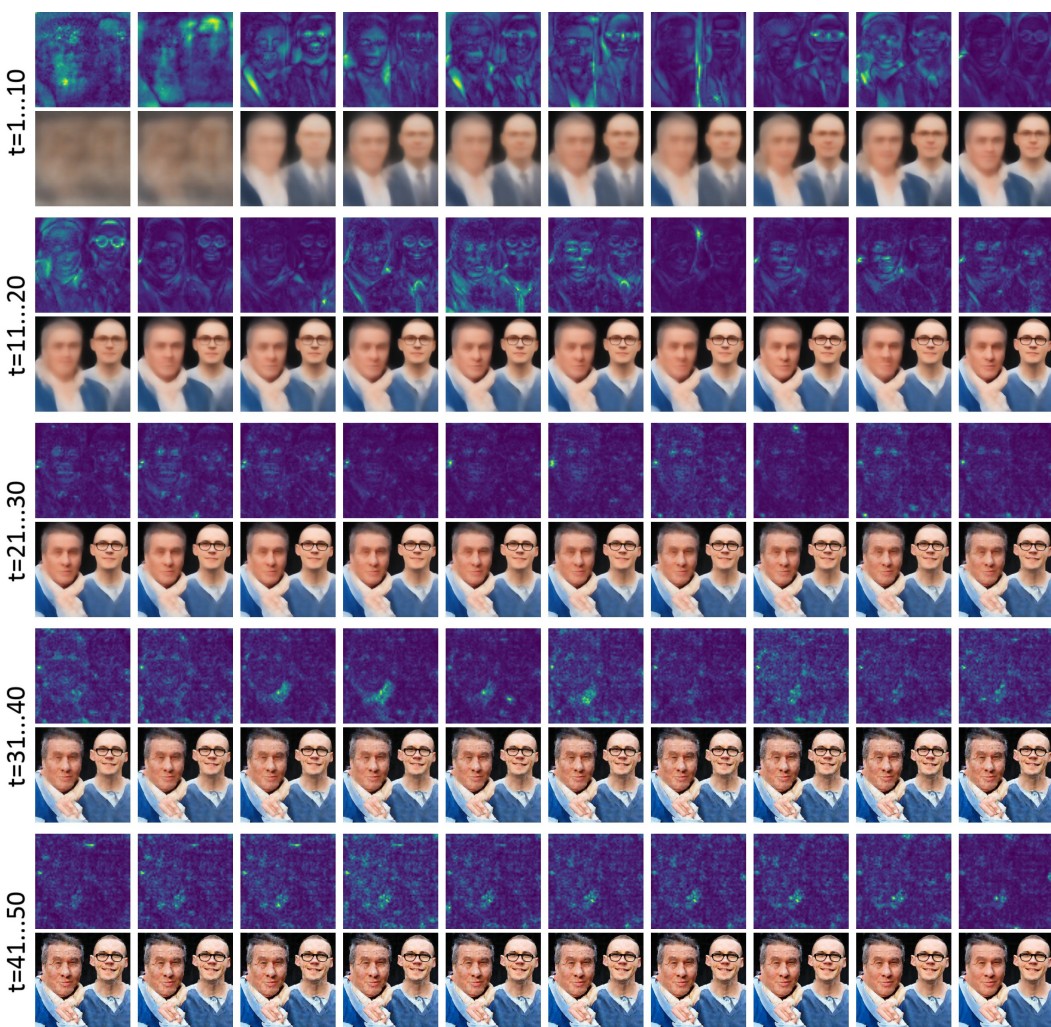

Figure S.9: **Example 1 of face voxel guided image synthesis for S1.** We utilize 50 steps of Multistep DPM-Solver++. We visualize the gradient magnitude w.r.t. the latent (top, normalized at each step for visualization) and the weighted euler RGB image that the brain encoder accepts (bottom).

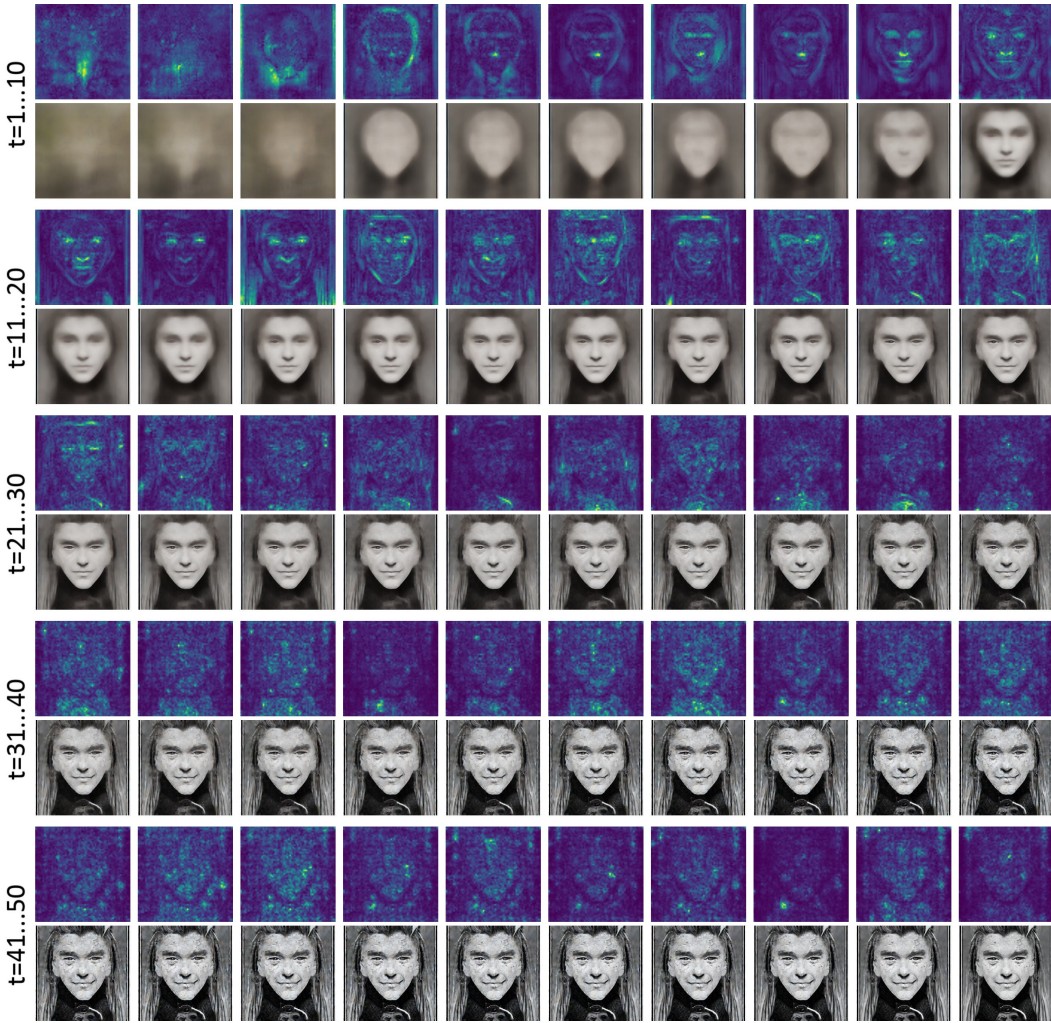

Figure S.10: **Example 2 of face voxel guided image synthesis for S1.** We utilize 50 steps of Multistep DPM-Solver++. We visualize the gradient magnitude w.r.t. the latent (top, normalized at each step for visualization) and the weighted euler RGB image that the brain encoder accepts (bottom)

## 10.2 Place voxels

We show examples where the end result contains an indoor scene (Figure S.11), or an outdoor scene (Figure S.12).

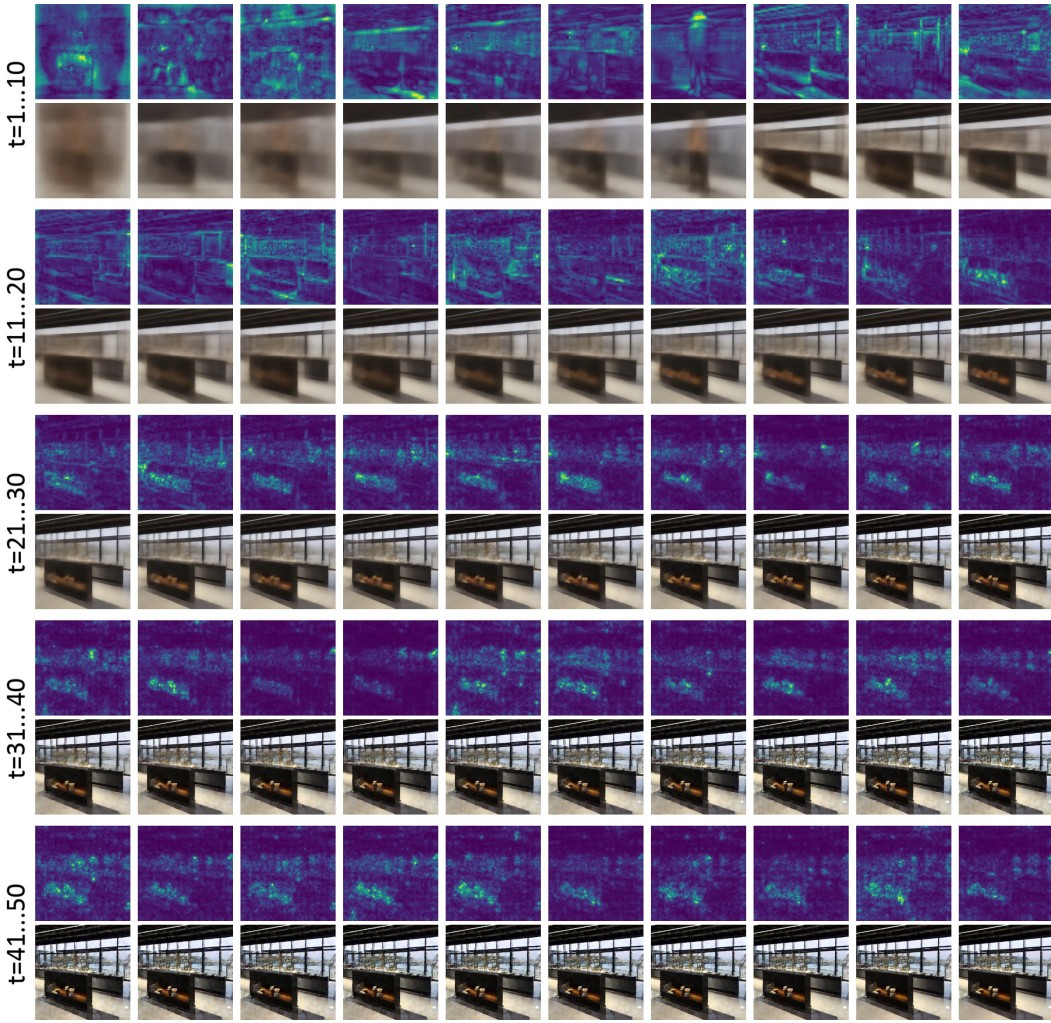

Figure S.11: **Example 1 of place voxel guided image synthesis for S1.** We utilize 50 steps of Multistep DPM-Solver++. We visualize the gradient magnitude w.r.t. the latent (top, normalized at each step for visualization) and the weighted euler RGB image that the brain encoder accepts (bottom).

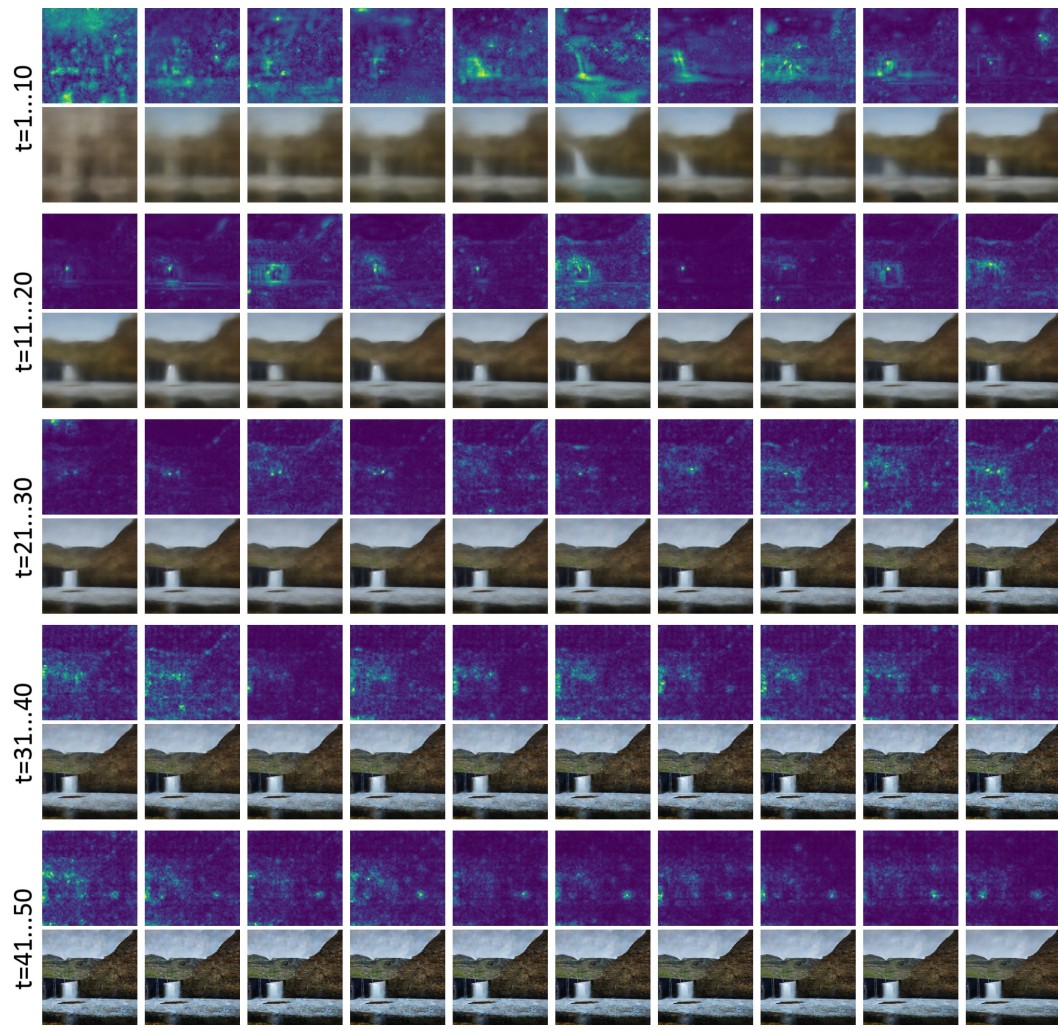

Figure S.12: **Example 2 of place voxel guided image synthesis for S1.** We utilize 50 steps of Multistep DPM-Solver++. We visualize the gradient magnitude w.r.t. the latent (top, normalized at each step for visualization) and the weighted euler RGB image that the brain encoder accepts (bottom).

## 10.3 Body voxels

We show examples where the end result contains an single person's body (Figure S.13), or an multiple people (Figure S.14).

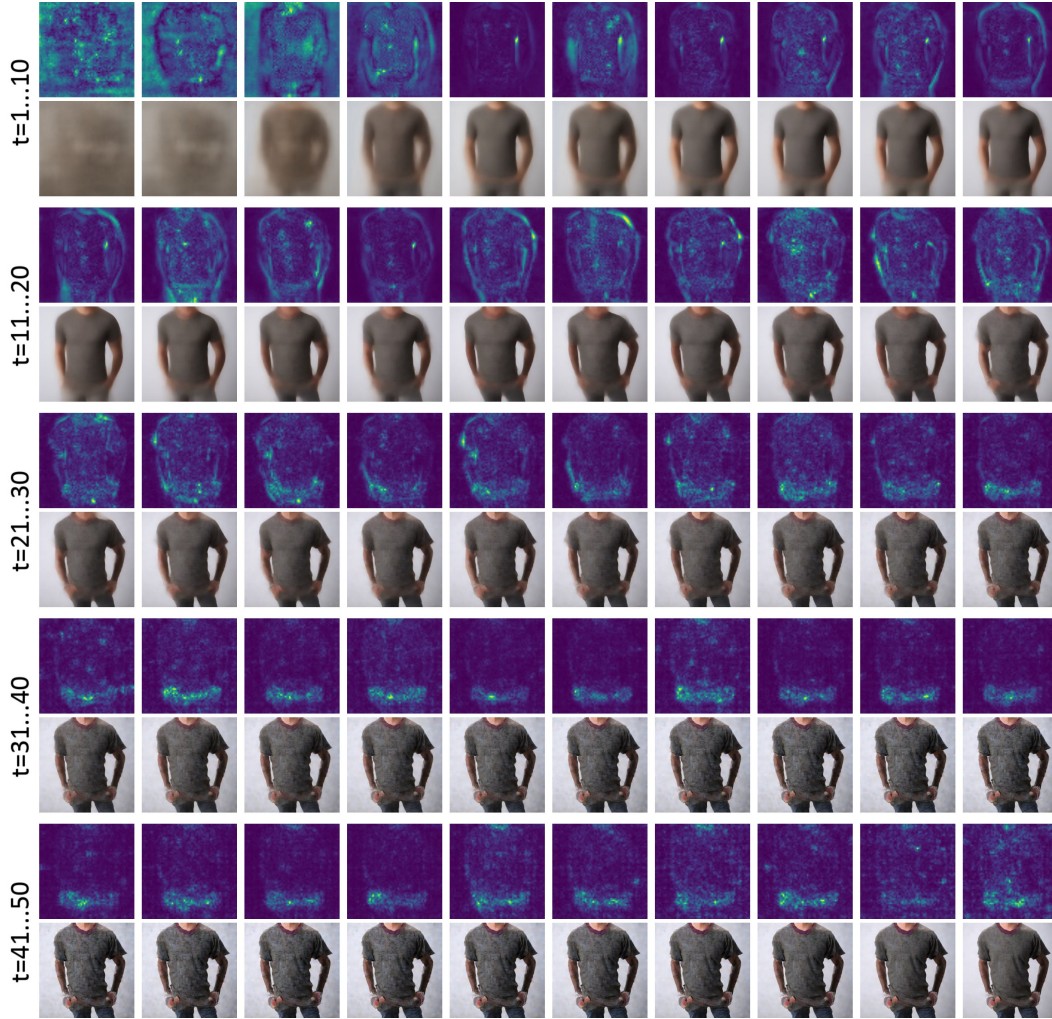

Figure S.13: **Example 1 of body voxel guided image synthesis for S1.** We utilize 50 steps of Multistep DPM-Solver++. We visualize the gradient magnitude w.r.t. the latent (top, normalized at each step for visualization) and the weighted euler RGB image that the brain encoder accepts (bottom).

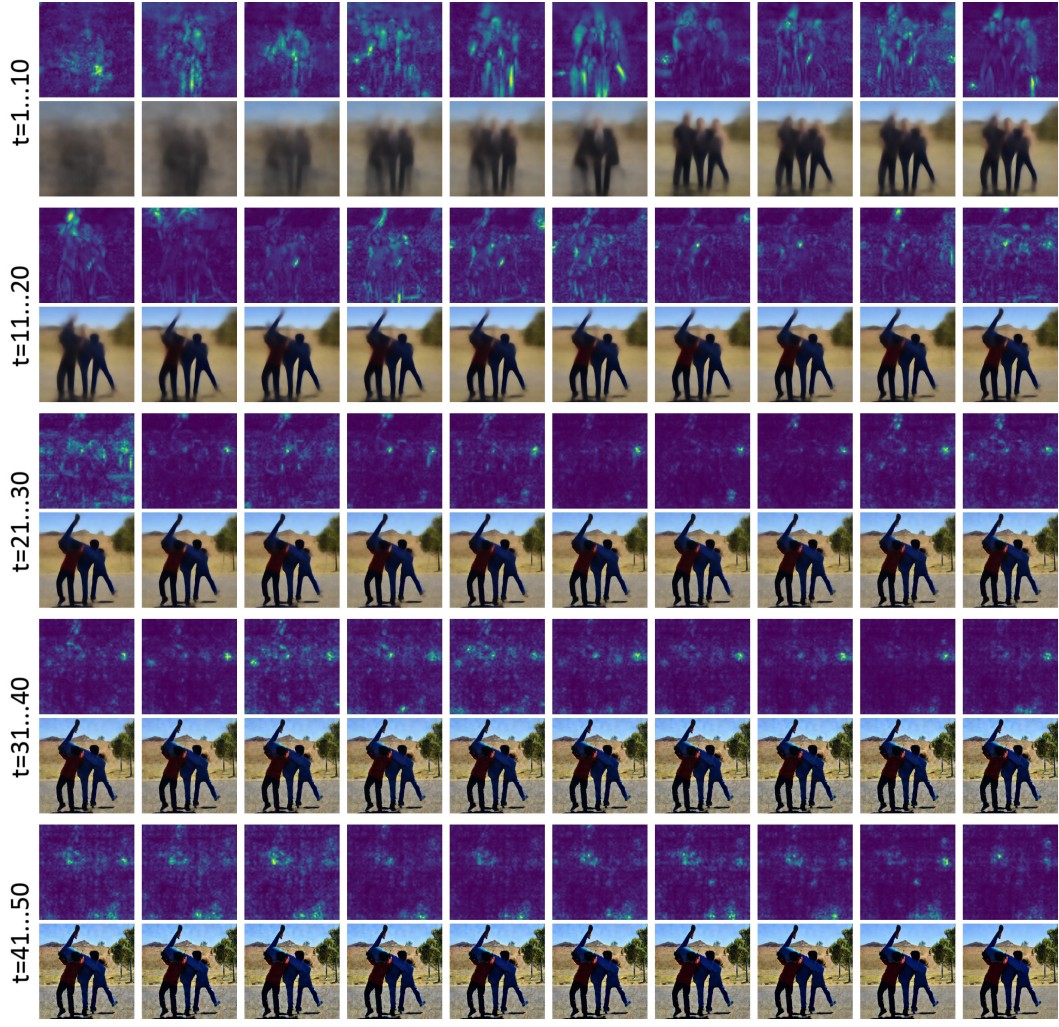

Figure S.14: **Example 2 of body voxel guided image synthesis for S1.** We utilize 50 steps of Multistep DPM-Solver++. We visualize the gradient magnitude w.r.t. the latent (top, normalized at each step for visualization) and the weighted euler RGB image that the brain encoder accepts (bottom).

## 10.4  Word voxels

We show examples where the end result contains recognizable words (Figure S.15), or glyph like objects (Figure S.16).

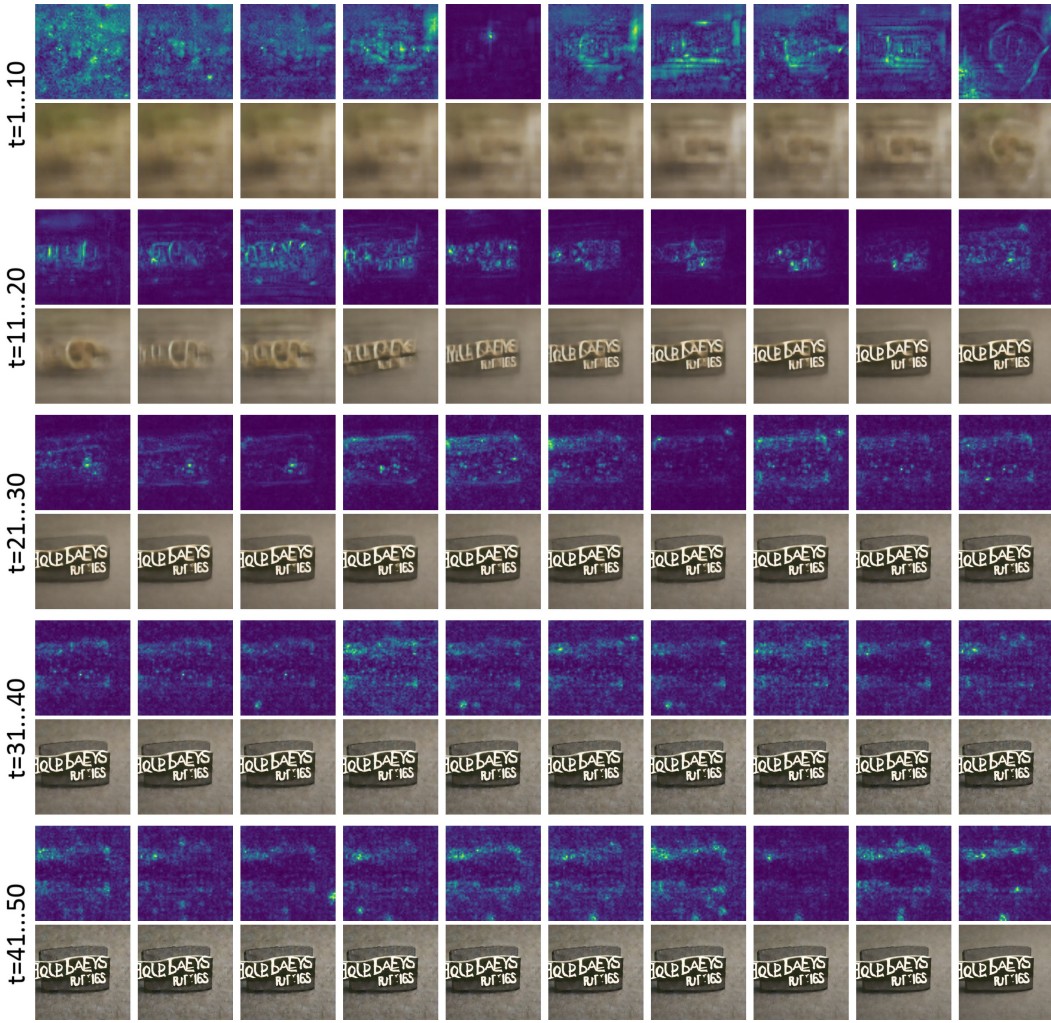

Figure S.15: **Example 1 of word voxel guided image synthesis for S1.** We utilize 50 steps of Multistep DPM-Solver++. We visualize the gradient magnitude w.r.t. the latent (top, normalized at each step for visualization) and the weighted euler RGB image that the brain encoder accepts (bottom).

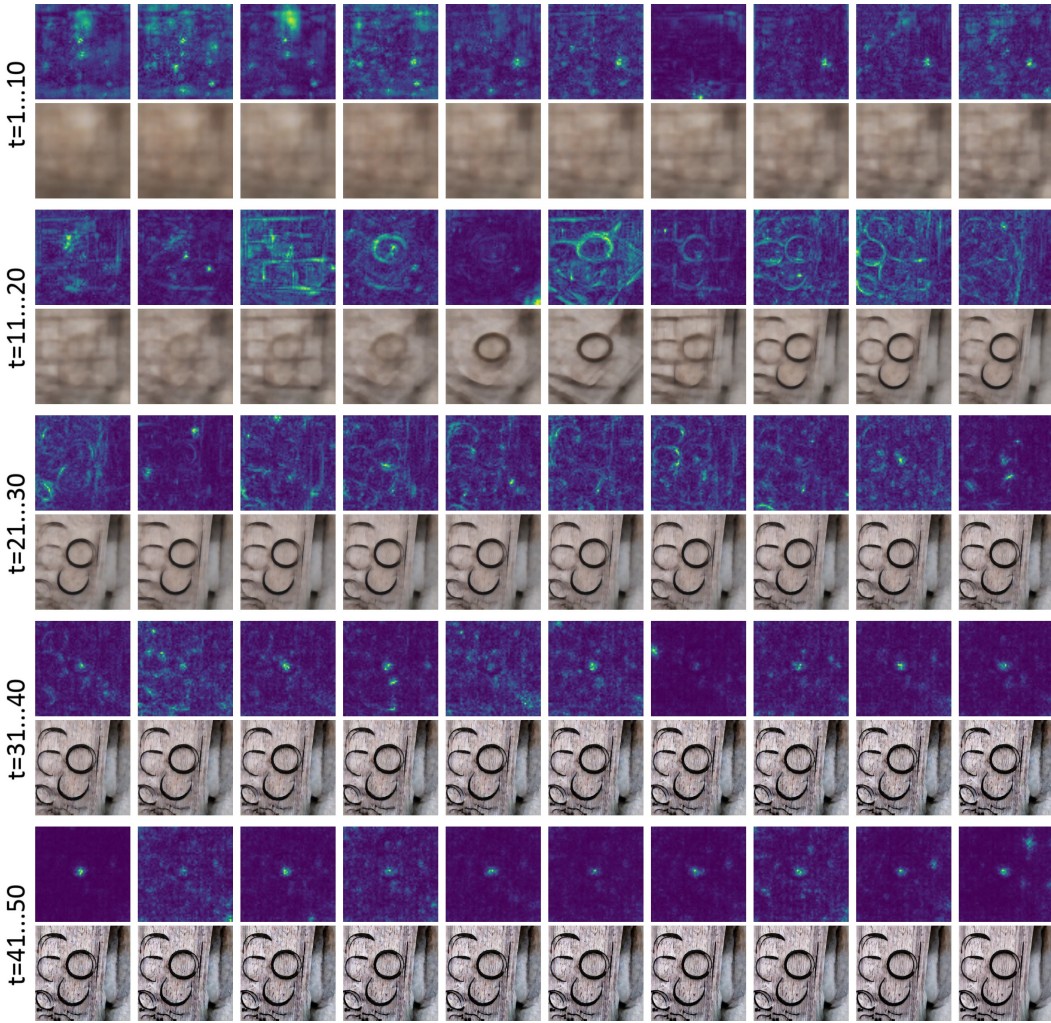

Figure S.16: **Example 2 of word voxel guided image synthesis for S1.** We utilize 50 steps of Multistep DPM-Solver++. We visualize the gradient magnitude w.r.t. the latent (top, normalized at each step for visualization) and the weighted euler RGB image that the brain encoder accepts (bottom).

## 10.5 Food voxels

We show examples where the end result contains highly processed foods (Figure S.17, showing what appears to be a cake), or cooked food containing vegetables (Figure S.18).

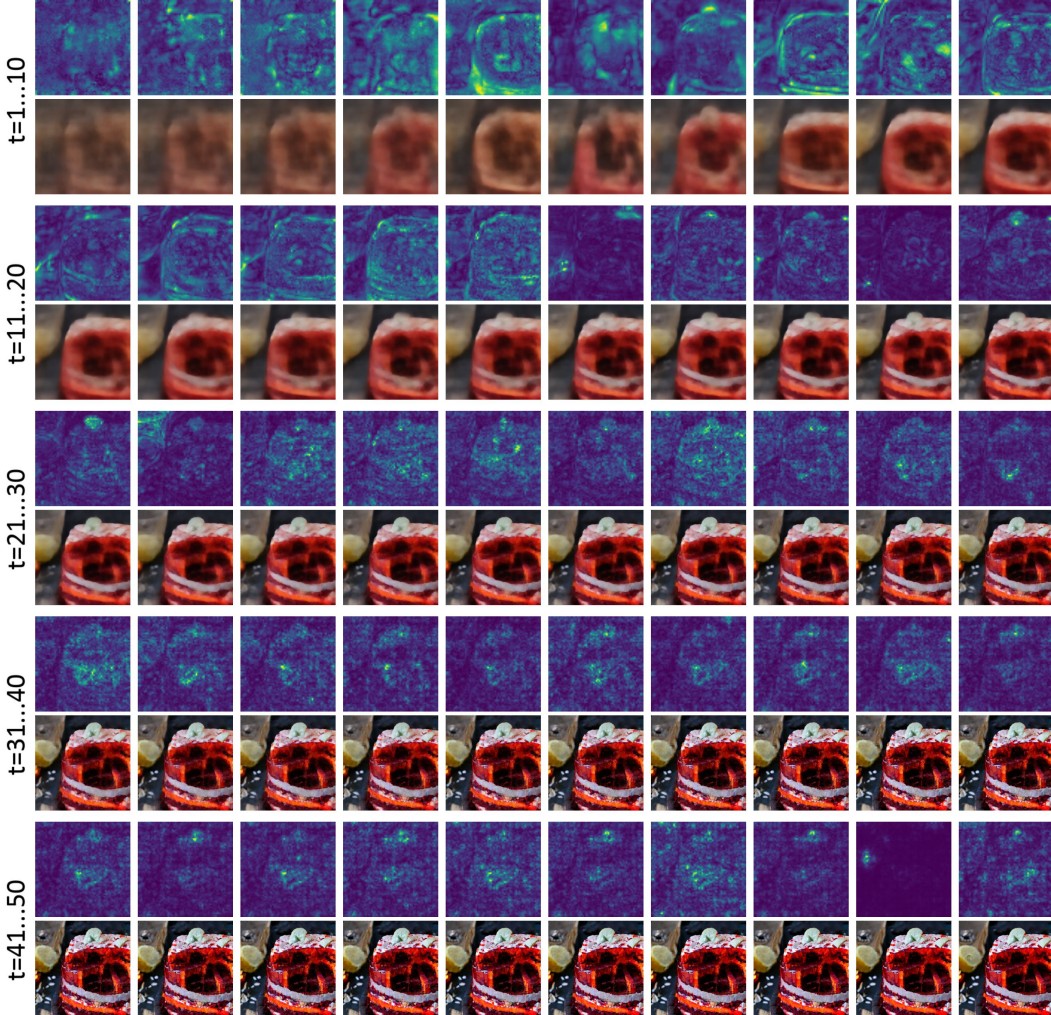

Figure S.17: **Example 1 of food voxel guided image synthesis for S1.** We utilize 50 steps of Multistep DPM-Solver++. We visualize the gradient magnitude w.r.t. the latent (top, normalized at each step for visualization) and the weighted euler RGB image that the brain encoder accepts (bottom).

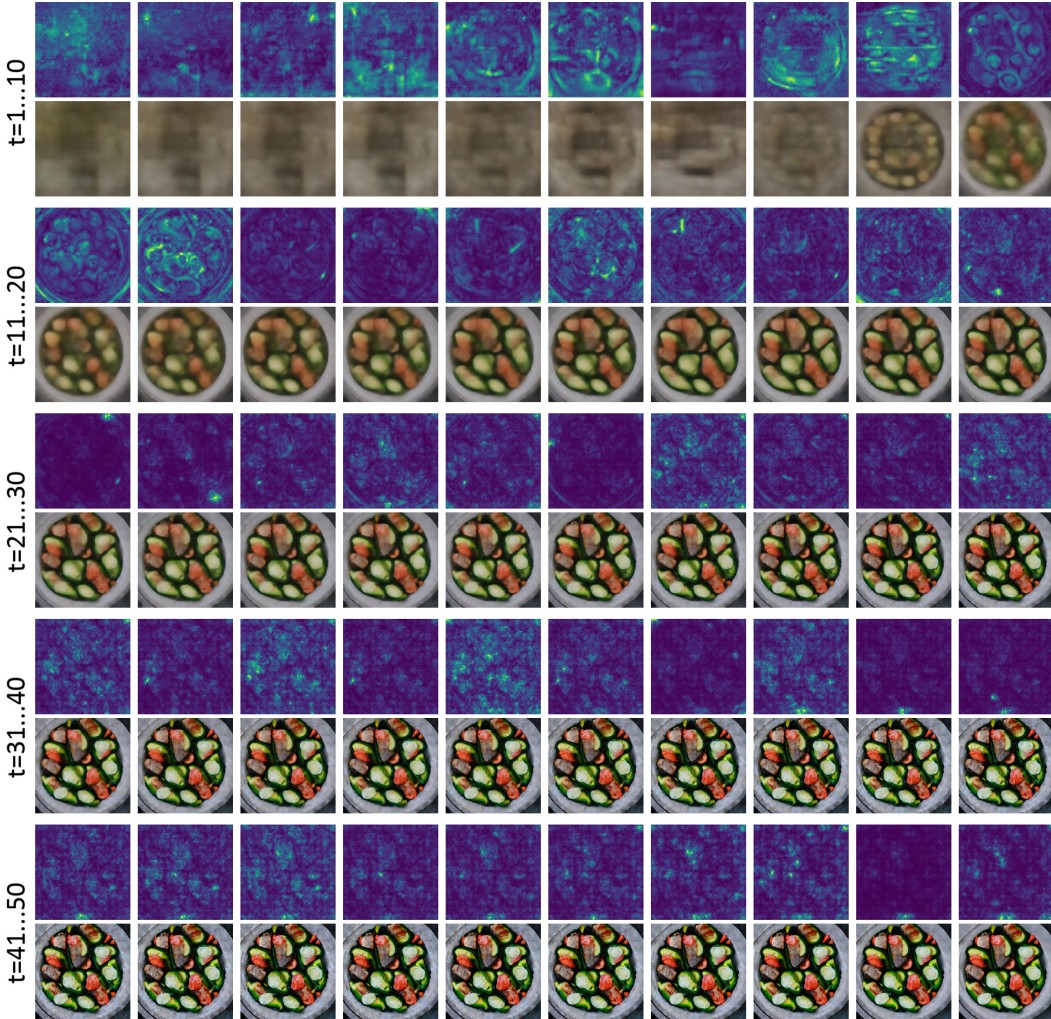

Figure S.18: **Example 2 of food voxel guided image synthesis for S1.** We utilize 50 steps of Multistep DPM-Solver++. We visualize the gradient magnitude w.r.t. the latent (top, normalized at each step for visualization) and the weighted euler RGB image that the brain encoder accepts (bottom).

# 11  Human behavioral study standard error

In this section, we show the human behavioral study results along with the standard error of the responses. Each question was answered by exactly 10 subjects from `prolific.co`. In each table, the results are show in the following format: **Mean(SEM)**. Where **Mean** is the average response, while **SEM** is the standard error of the mean ratio across 10 subjects: (SEM $= \frac{\sigma}{\sqrt{10}}$).

| Which ROI has more... | photorealistic faces | | | | animals | | | | abstract shapes/lines | | | |
|---|---|---|---|---|---|---|---|---|---|---|---|---|
| | S1 | S2 | S5 | S7 | S1 | S2 | S5 | S7 | S1 | S2 | S5 | S7 |
| FFA-NSD | **45**(7.2) | **43**(8.3) | **34**(6.2) | **41**(6.5) | 34(4.5) | 34(3.5) | 17(4.0) | 15(3.8) | 21(6.8) | 6(4.0) | 14(2.9) | 22(6.6) |
| OFA-NSD | 25(5.1) | 22(6.4) | 21(5.6) | 18(5.3) | **47**(3.2) | **36**(2.5) | **65**(5.7) | **65**(6.4) | **24**(8.5) | **44**(9.2) | **28**(8.1) | **25**(6.4) |
| FFA-BrainDiVE | **79**(7.8) | **89**(4.8) | **60**(5.3) | **52**(5.3) | 17(5.6) | 13(3.5) | 21(3.9) | 19(2.2) | 6(3.2) | 11(6.4) | 18(4.9) | 20(6.6) |
| OFA-BrainDiVE | 11(5.7) | 4(2.5) | 15(2.9) | 22(5.1) | **71**(8.4) | **61**(8.2) | **52**(5.1) | **50**(3.5) | **80**(5.8) | **79**(7.4) | **40**(5.8) | **39**(7.1) |

Table S.5: **Human evaluation of the difference between face-selective ROIs**. Evaluators compare groups of images corresponding to OFA and FFA; comparisons are done within GT and generated images respectively. Questions are posed as: "Which group of images has more X?"; options are FFA/OFA/Same. Results are in %. Note that the "Same" responses are not shown; responses across all three options sum to 100.

| Which cluster is more... | vegetables/fruits | | | | healthy | | | |
|---|---|---|---|---|---|---|---|---|
| | S1 | S2 | S5 | S7 | S1 | S2 | S5 | S7 |
| Food-1 NSD | 17(4.3) | 21(4.8) | 27(5.1) | 36(3.5) | 28(5.8) | 22(3.7) | 29(6.2) | 40(4.0) |
| Food-2 NSD | **65**(7.2) | **56**(6.4) | **56**(5.7) | **49**(3.9) | **50**(7.1) | **47**(4.9) | **54**(6.0) | **45**(4.3) |
| Food-1 BrainDiVE | 11(7.0) | 10(6.0) | 8(6.6) | 11(6.5) | 15(6.2) | 16(6.0) | 20(7.2) | 17(7.1) |
| Food-2 BrainDiVE | **80**(7.3) | **75**(8.0) | **67**(9.8) | **64**(7.4) | **68**(7.7) | **68**(7.3) | **46**(9.3) | **51**(7.8) |

| Which cluster is more... | colorful | | | | far away | | | |
|---|---|---|---|---|---|---|---|---|
| | S1 | S2 | S5 | S7 | S1 | S2 | S5 | S7 |
| Food-1 NSD | 19(5.5) | 18(6.1) | 13(2.8) | 27(3.5) | 32(6.6) | 24(4.7) | 23(6.5) | 28(4.2) |
| Food-2 NSD | **42**(6.4) | **52**(5.6) | **53**(6.5) | **42**(6.4) | **34**(7.0) | **39**(8.1) | **36**(7.9) | **42**(7.3) |
| Food-1 BrainDiVE | 6(3.8) | 9(5.7) | 11(5.7) | 16(4.9) | 24(6.8) | 18(6.4) | 27(8.9) | 18(6.0) |
| Food-2 BrainDiVE | **79**(7.9) | **82**(6.9) | **65**(7.6) | **61**(8.9) | **39**(10.1) | **51**(9.0) | **39**(8.8) | **40**(8.8) |

Table S.6: **Human evaluation of the difference between food clusters**. Evaluators compare groups of images corresponding to food cluster 1 (Food-1) and food cluster 2 (Food-2), with questions posed as "Which group of images has/is more X?". Comparisons are done within NSD and generated images respectively. Note that the "Same" responses are not shown; responses across all three options sum to 100. Results are in %.

| Which cluster is more... | angular/geometric | | | | indoor | | | |
|---|---|---|---|---|---|---|---|---|
| | S1 | S2 | S5 | S7 | S1 | S2 | S5 | S7 |
| OPA-1 NSD | **45**(7.2) | **58**(9.4) | **49**(7.7) | **51**(9.0) | **71**(5.6) | **88**(4.9) | **80**(5.1) | **79**(5.6) |
| OPA-2 NSD | 13(4.0) | 12(2.4) | 14(2.9) | 16(4.5) | 7(3.2) | 8(3.7) | 11(3.0) | 14(4.3) |
| OPA-1 BrainDiVE | **76**(7.8) | **87**(8.6) | **88**(6.6) | **76**(7.8) | **89**(5.6) | **90**(5.7) | **90**(4.7) | **85**(5.3) |
| OPA-2 BrainDiVE | 12(4.9) | 3(2.0) | 4(1.5) | 10(4.2) | 7(3.2) | 7(3.2) | 5(2.1) | 8(2.4) |

| Which cluster is more... | natural | | | | far away | | | |
|---|---|---|---|---|---|---|---|---|
| | S1 | S2 | S5 | S7 | S1 | S2 | S5 | S7 |
| OPA-1 NSD | 14(3.8) | 3(2.0) | 9(4.1) | 10(2.8) | 10(2.4) | 1(0.9) | 6(2.9) | 8(2.4) |
| OPA-2 NSD | **73**(3.4) | **89**(7.4) | **71**(6.4) | **81**(6.1) | **69**(4.6) | **93**(3.8) | **81**(6.5) | **85**(5.5) |
| OPA-1 BrainDiVE | 6(3.2) | 6(1.5) | 9(3.6) | 6(2,9) | 1(0.9) | 3(2.8) | 3(2.8) | 8(5.6) |
| OPA-2 BrainDiVE | **91**(5.7) | **91**(3.6) | **83**(6.9) | **90**(5.5) | **97**(2.8) | **92**(6.6) | **91**(5.6) | **88**(7.4) |

Table S.7: **Human evaluation of the difference between OPA clusters**. Evaluators compare groups of images corresponding to OPA cluster 1 (OPA-1) and OPA cluster 2 (OPA-2), with questions posed as "Which group of images is more X?". Comparisons are done within NSD and generated images respectively. Note that the "Same" responses are not shown; responses across all three options sum to 100. Results are in %.

# 12 Brain encoder $R^2$

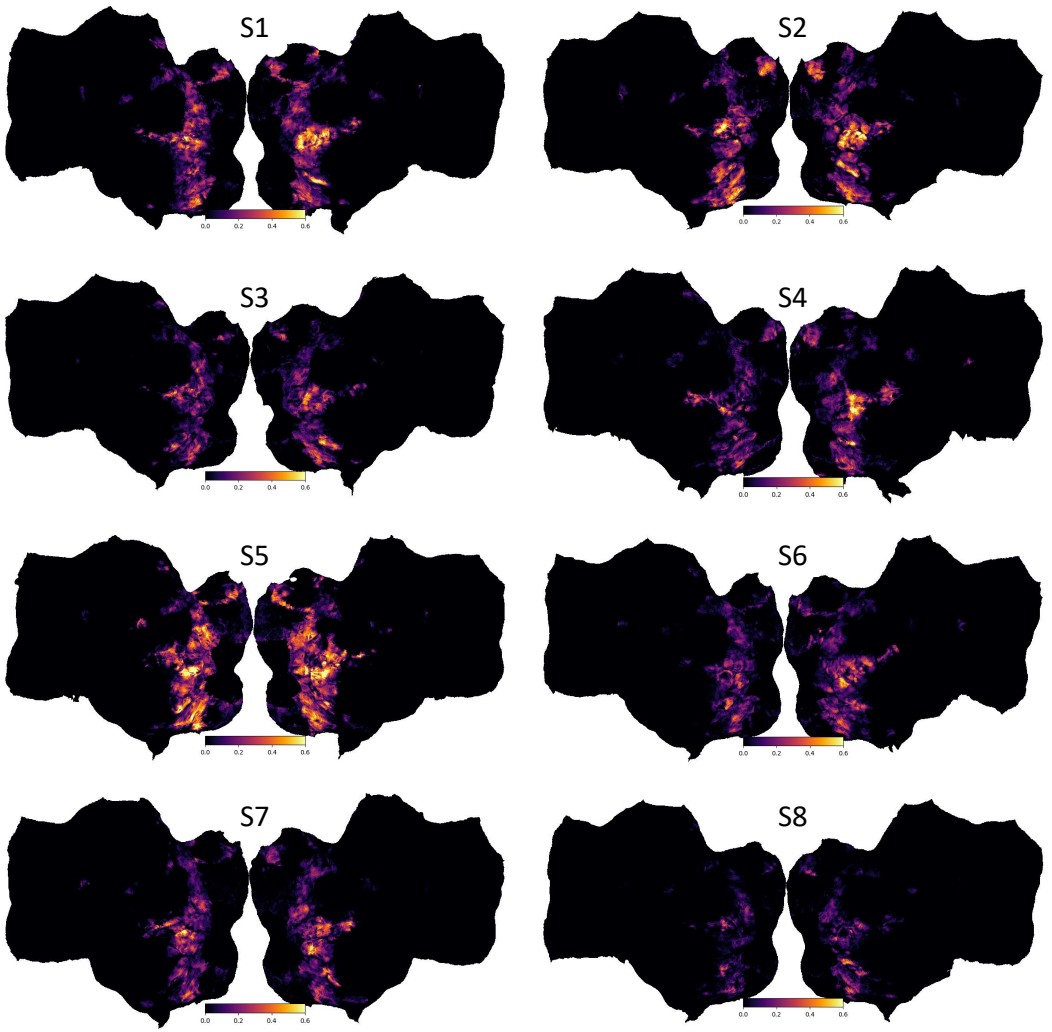

Figure S.19: **Visualization of $R^2$ on test set images.** We evaluate $R^2$ on the $\sim 1000$ images shared by all subjects. Note that voxels in early visual or outside of higher visual are not modeled.

In Figure 12 we show the $R^2$ of the brain encoder as evaluated on the test images. Our brain encoder consists of a CLIP backbone and a linear adaptation layer. We do not model voxels in the early visual cortex, nor do we model voxels outside of higher visual. Our model can generally achieve high $R^2$ in regions in known regions of visual semantic selectivity.

# 13 OFA and FFA visualizations

In this section, we visualize the top-10 NSD and BrainDiVE images for OFA and FFA. NSD images are selected using the fMRI betas averaged within each ROI. BrainDiVE images are ranked using our predicted ROI activities from 500 images.

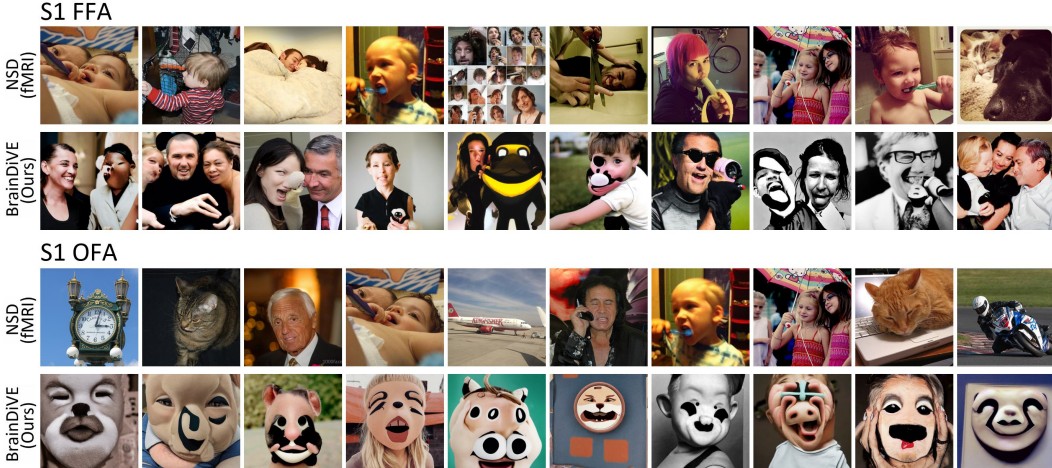

Figure S.20: **Results for face-selective ROIs in S1.**

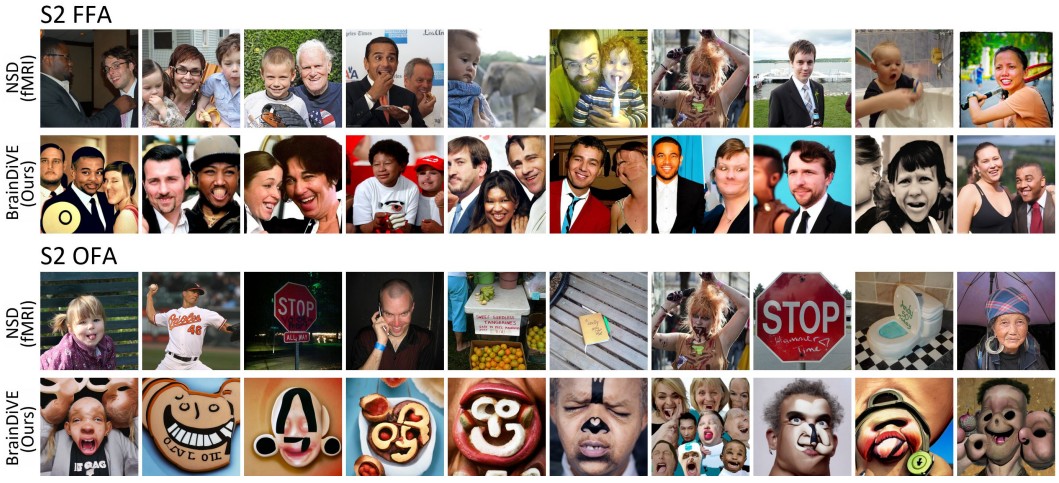

Figure S.21: **Results for face-selective ROIs in S2.**

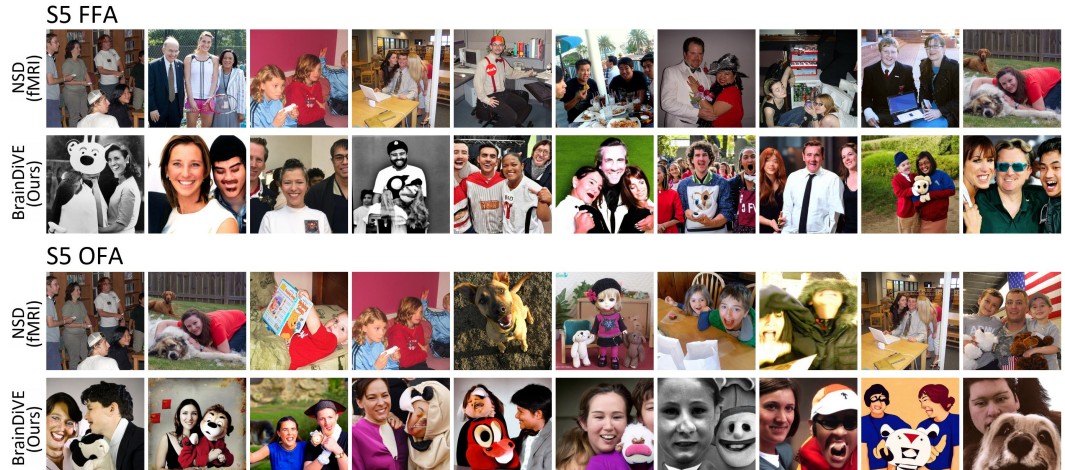

Figure S.22: **Results for face-selective ROIs in S5.**

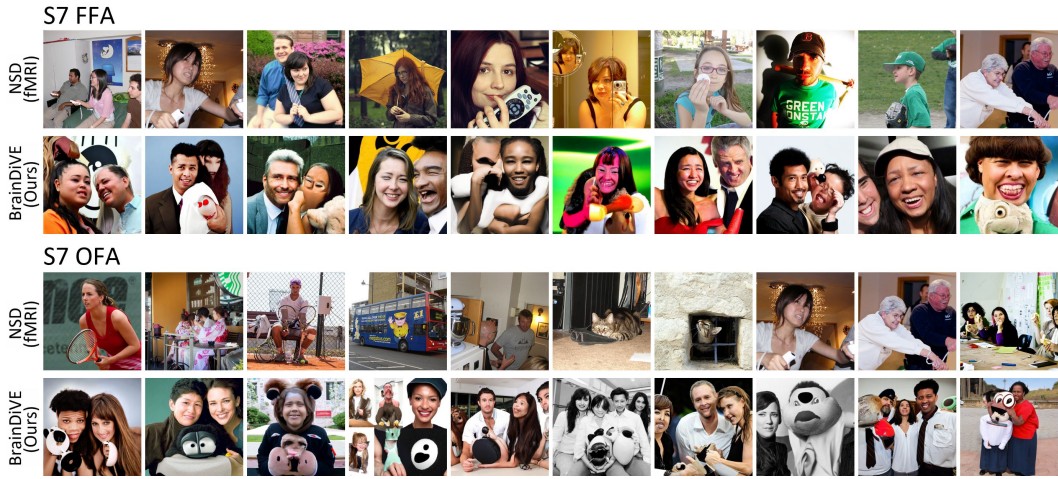

Figure S.23: **Results for face-selective ROIs in S7.**

# 14 OPA and food visualizations

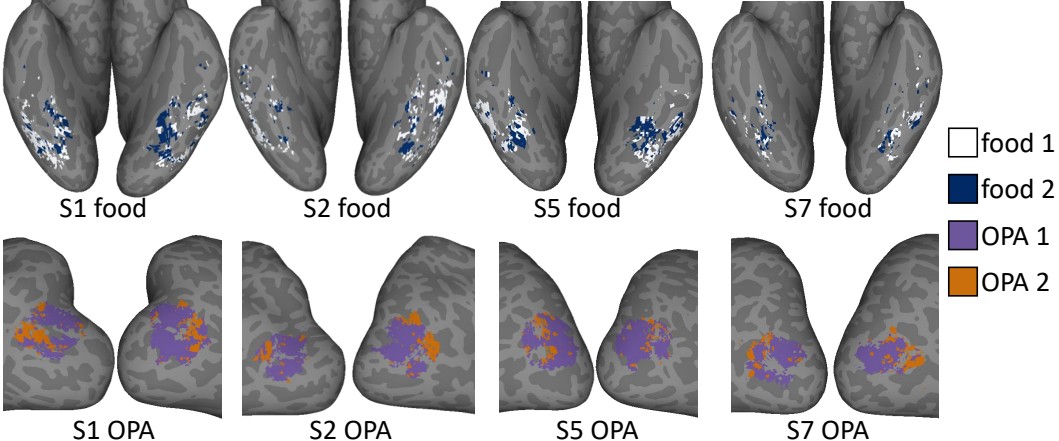

Figure S.24: **Clustering within the food ROI and within OPA.** Clustering of encoder model weights for each region is shown for four subjects on an inflated cortical surface.

Consistent with Jain et al. [5], we observe that the food voxels themselves are anatomically variable across subjects, while the two food clusters form alternating patches within the food patches. OPA generally yields anatomically consistent clusters in the four subjects we investigated, with all four subjects showing an anterior-posterior split for OPA.

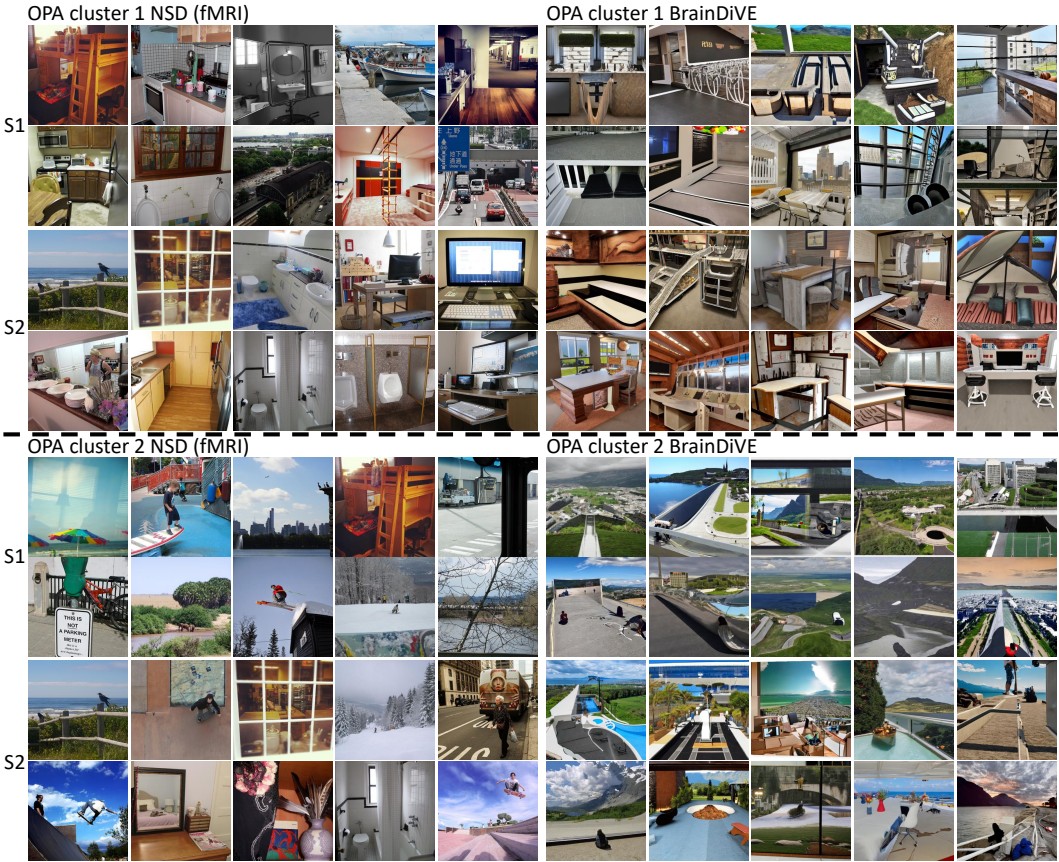

Figure S.25: **Comparing results across the OPA clusters for S1 and S2.**

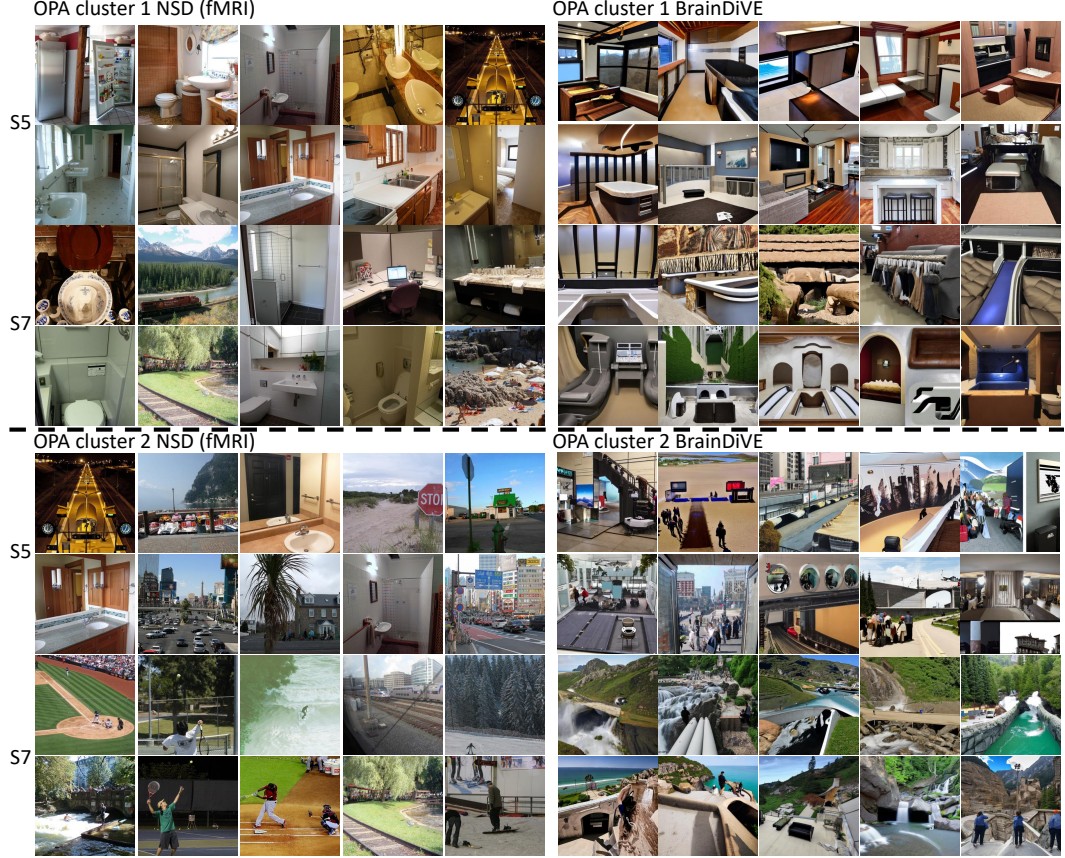

Figure S.26: **Comparing results across the OPA clusters for S5 and S7.**

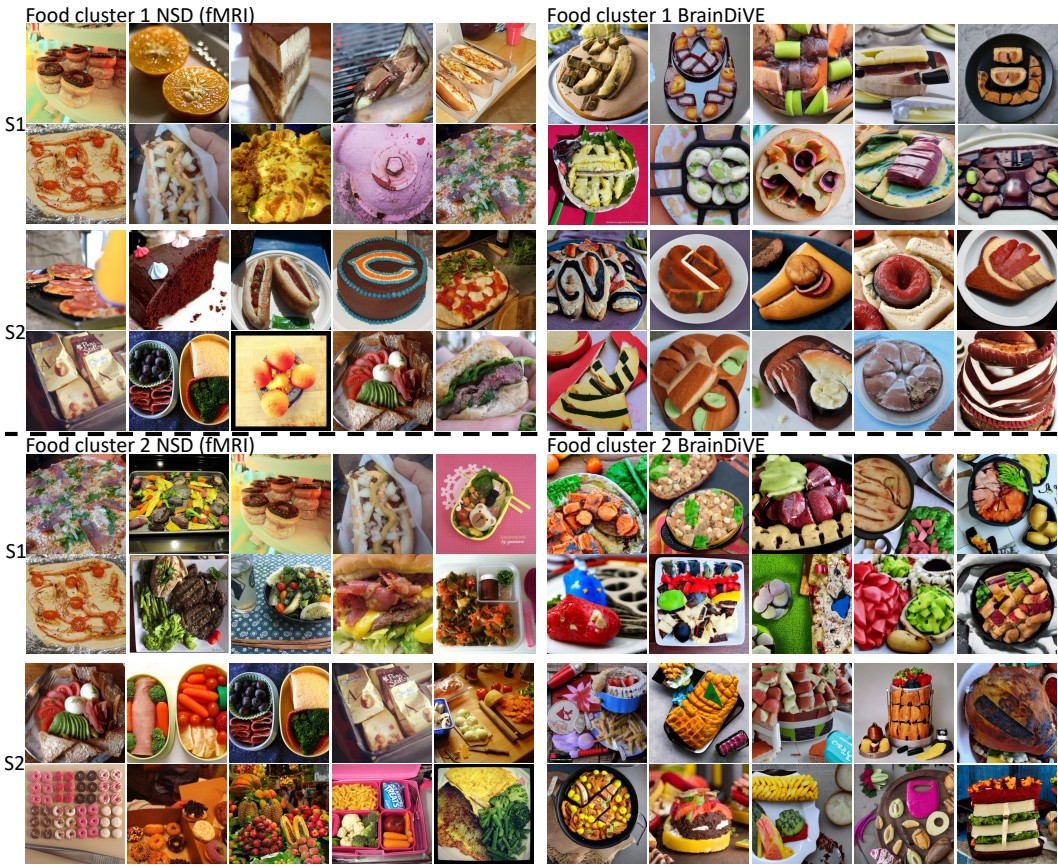

Figure S.27: **Comparing results across the food clusters for S1 and S2.**

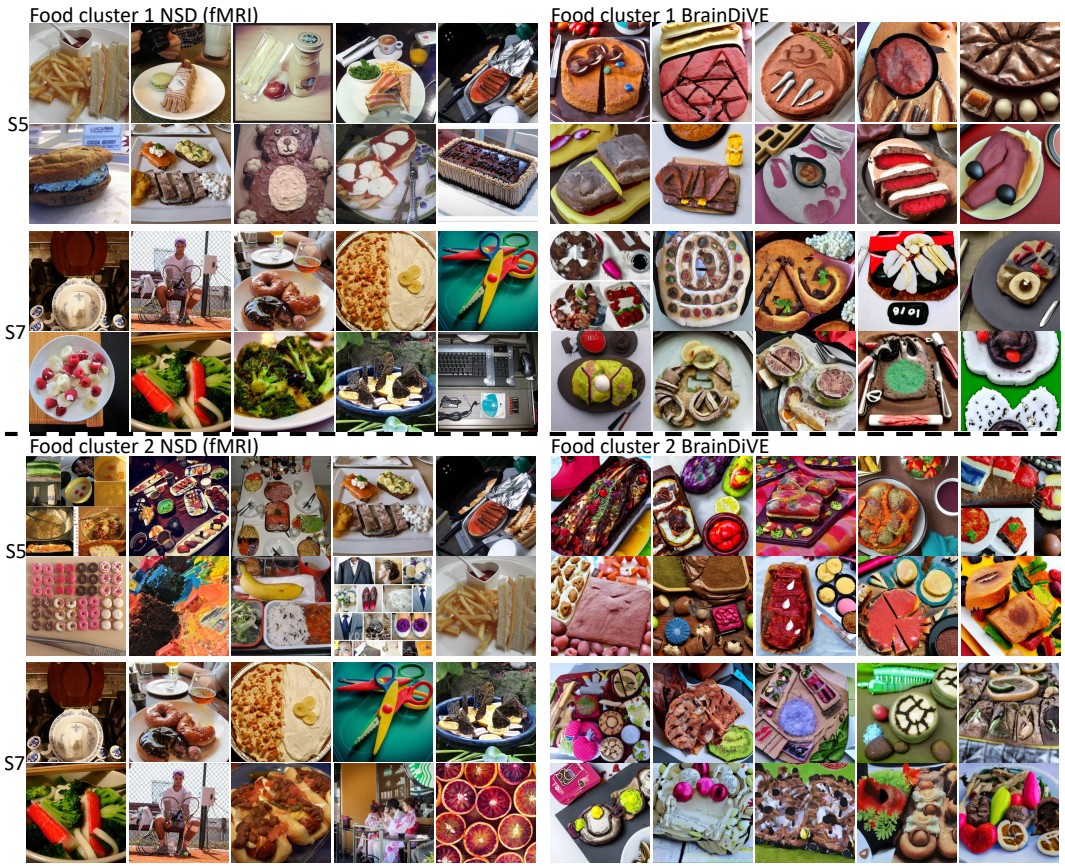

Figure S.28: **Comparing results across the food clusters for S5 and S7.**

## 15 Training, inference, and compute details

**Encoder training.** Our encoder backbone uses `ViT-B/16` with CLIP pretrained weights `laion2b_s34b_b88k` provided by OpenCLIP [65, 66]. The ViT [74] weights for the brain encoder are frozen. We train a linear layer consisting of weight and bias to map from the 512 dimensional vector to higher visual voxels $B$. The CLIP image branch outputs are normalized to the unit sphere.

$$M_\theta(\mathcal{I}) = W \times \frac{\texttt{CLIP}_{\texttt{img}}(\mathcal{I})}{\|\texttt{CLIP}_{\texttt{img}}(\mathcal{I})\|_2} + b$$

Training is done using the `Adam` optimizer [75] with learning rate $\texttt{lr}_{\texttt{init}} = 3e - 4$ and $\texttt{lr}_{\texttt{end}} = 1.5e - 4$, with learning rate adjusting exponentially each epoch. We train for 100 epochs. Decoupled weight decay [76] of magnitude $\texttt{decay} = 2e - 2$ is applied. Each subject is trained independently using the $\sim 9000$ images unique to each subject's stimulus set, with $R^2$ evaluated on the $\sim 1000$ images shared by all subjects.

During training of the encoder weights, the image is resized to $224 \times 224$ to match the input size of `ViT-B/16`. We augment the images by first randomly scaling the pixels by a value between $[0.95, 1.05]$, then normalize the image using OpenCLIP ViT image mean and variance. Prior to input to the network, we further randomly offset the image spatially by up to 4 pixels along the height and width dimensions. The empty pixels are filled in using edge value padding. A small amount of gaussian noise $\mathcal{N}(0, 0.05^2)$ is added to each pixel prior to input to the encoder backbone.

**Objective.** For all experiments, the objective used is the maximization of a selected set of voxels. Here we will further draw a link between the optimization objective we use and the traditional CLIP text prompt guidance objective [14, 59]. Recall that $M_\theta$ is our brain activation encoder that maps from the image to per-voxel activations. It accepts as input an image, passes it through a ViT backbone, normalizes that vector to the unit sphere, then applies a linear mapping to go to per-voxel activations. $S \in N$ are the set of voxels we are currently trying to maximize (where $N$ is the set of all voxels in the brain), $\gamma$ is a step size parameter, and $D_\Omega$ is the decoder from the latent diffusion model that outputs an RGB image (we ignore the euler approximation for clarity). Also recall that we use a diffusion model that performs $\epsilon$-prediction.

In the general case, we perturb the denoising process by trying to maximize a set of voxels $S$:

$$\epsilon'_{theta} = \epsilon_{theta} - \sqrt{1 - \alpha_t} \nabla_{x_t} \left( \frac{\gamma}{|S|} \sum_{i \in S} M_\theta(D_\Omega(x'_t))_i \right)$$

For the purpose of this section, we will focus on a single voxel first, then discuss the multi-voxel objective.

In our case, the single voxel perturbation is (assuming $W$ is a vector, and that $\langle \cdot, \cdot \rangle$ is the inner product):

$$\epsilon'_{theta} = \epsilon_{theta} - \sqrt{1 - \alpha_t} \nabla_{x_t} (\gamma M_\theta(D_\Omega(x'_t)))$$
$$= \epsilon_{theta} - \sqrt{1 - \alpha_t} \nabla_{x_t} (\gamma M_\theta(\mathcal{I}_{\text{gen}}))$$
$$= \epsilon_{theta} - \gamma \sqrt{1 - \alpha_t} \nabla_{x_t} \left( \left\langle W, \frac{\texttt{CLIP}_{\texttt{img}}(\mathcal{I}_{\text{gen}})}{\|\texttt{CLIP}_{\texttt{img}}(\mathcal{I}_{\text{gen}})\|_2} \right\rangle + b \right)$$

We can ignore $b$, as it does not affect optimal $\texttt{CLIP}_{\texttt{img}}(\mathcal{I}_{\text{gen}})$

$$\equiv \epsilon_{theta} - \gamma \sqrt{1 - \alpha_t} \nabla_{x_t} \left( \left\langle W, \frac{\texttt{CLIP}_{\texttt{img}}(\mathcal{I}_{\text{gen}})}{\|\texttt{CLIP}_{\texttt{img}}(\mathcal{I}_{\text{gen}})\|_2} \right\rangle \right)$$

Now let us consider the typical CLIP guidance objective for diffusion models, where $\mathcal{P}_{\text{text}}$ is the guidance prompt, and $\texttt{CLIP}_{\texttt{text}}$ is the text encoder component of CLIP:

$$\epsilon'_{theta} = \epsilon_{theta} - \gamma \sqrt{1 - \alpha_t} \nabla_{x_t} \left( \left\langle \frac{\texttt{CLIP}_{\texttt{text}}(\mathcal{P}_{\text{text}})}{\|\texttt{CLIP}_{\texttt{text}}(\mathcal{P}_{\text{text}})\|_2}, \frac{\texttt{CLIP}_{\texttt{img}}(\mathcal{I}_{\text{gen}})}{\|\texttt{CLIP}_{\texttt{img}}(\mathcal{I}_{\text{gen}})\|_2} \right\rangle \right)$$

As such, the $W$ that we find by linearly fitting CLIP image embeddings to brain activation plays the role of a text prompt. In reality, $\|W\|_2 \neq 1$ (but norm is a constant for each voxel), and there is likely

no computationally efficient way to "invert" $W$ directly into a human interpretable text prompt. By performing brain guidance, we are essentially using the diffusion model to synthesize an image $\mathcal{I}_{\text{gen}}$ where in addition to satisfying the natural image constraint, the image also attempts to satisfy:

$$\frac{\text{CLIP}_{\text{img}}(\mathcal{I}_{\text{gen}})}{\|\text{CLIP}_{\text{img}}(\mathcal{I}_{\text{gen}})\|_2} = \frac{W}{\|W\|_2}$$

Or put another way, it generates images where the CLIP latent is aligned with the direction of $W$. Let us now consider the multi-voxel perturbation, where $W_i, b_i$ is the per-voxel weight vector and bias:

$$\epsilon'_{theta} = \epsilon_{theta} - \sqrt{1-\alpha_t}\nabla_{x_t}(\frac{\gamma}{|S|}\sum_{i \in S} M_\theta(D_\Omega(x'_t))_i)$$

We move $\frac{\gamma}{|S|}$ outside of the gradient operation

$$= \epsilon_{theta} - \frac{\gamma}{|S|}\sqrt{1-\alpha_t}\nabla_{x_t}(\sum_{i \in S} M_\theta(D_\Omega(x'_t))_i)$$

$$= \epsilon_{theta} - \frac{\gamma}{|S|}\sqrt{1-\alpha_t}\nabla_{x_t}(\sum_{i \in S} \left[\langle W_i, \frac{\text{CLIP}_{\text{img}}(\mathcal{I}_{\text{gen}})}{\|\text{CLIP}_{\text{img}}(\mathcal{I}_{\text{gen}})\|_2}\rangle + b_i\right])$$

We again ignore $b_i$ as it does not affect gradient

$$\equiv \epsilon_{theta} - \frac{\gamma}{|S|}\sqrt{1-\alpha_t}\nabla_{x_t}(\sum_{i \in S}\langle W_i, \frac{\text{CLIP}_{\text{img}}(\mathcal{I}_{\text{gen}})}{\|\text{CLIP}_{\text{img}}(\mathcal{I}_{\text{gen}})\|_2}\rangle)$$

We can move $\sum$ outside due to the distributive nature of gradients

$$= \epsilon_{theta} - \frac{\gamma}{|S|}\sqrt{1-\alpha_t}\sum_{i \in S}\left[\nabla_{x_t}(\langle W_i, \frac{\text{CLIP}_{\text{img}}(\mathcal{I}_{\text{gen}})}{\|\text{CLIP}_{\text{img}}(\mathcal{I}_{\text{gen}})\|_2}\rangle)\right]$$

Thus from a gradient perspective, the total gradient is the average of gradients from all voxels. Recall that the inner product is a bilinear function, and that the CLIP image latent is on the unit sphere. Then we are generating an image that

$$\frac{\text{CLIP}_{\text{img}}(\mathcal{I}_{\text{gen}})}{\|\text{CLIP}_{\text{img}}(\mathcal{I}_{\text{gen}})\|_2} = \frac{\sum_{i \in S} W_i}{\|\sum_{i \in S} W_i\|_2}$$

Where the optimal image has a CLIP latent that is aligned with the direction of $\sum_{i \in S} W_i$.

**Compute.** We perform our experiments on a cluster of `Nvidia V100` GPUs in either 16GB or 32GB VRAM configuration, and all experiments consumed approximately $1,500$ compute hours. Each image takes between 20 and 30 seconds to synthesize. All experiments were performed using PyTorch, with cortex visualizations done using PyCortex [77].

**CLIP prompts.** Here we list the text prompts that are used to classify the images for Table 1. in the main paper.

```
face_class = ["A face facing the camera", "A photo of a face", "A photo of
a human face", "A photo of faces", "A photo of a person's face", "A person
looking at the camera", "People looking at the camera","A portrait of a
person", "A portrait photo"]

body_class = ["A photo of a torso", "A photo of torsos", "A photo of limbs",
"A photo of bodies", "A photo of a person", "A photo of people"]

scene_class = ["A photo of a bedroom", "A photo of an office","A photo of a
hallway", "A photo of a doorway", "A photo of interior design", "A photo
```

```
of a building", "A photo of a house", "A photo of nature", "A photo of
landscape", "A landscape photo", "A photo of trees", "A photo of grass"]

food_class = ["A photo of food"]

text_class = ["A photo of words", "A photo of glyphs", "A photo of a glyph",
"A photo of text", "A photo of numbers", "A photo of a letter", "A photo of
letters", "A photo of writing", "A photo of text on an object"]
```

We classify an image as belonging to a category if the image's CLIP latent has highest cosine similarity with the CLIP latent of a prompt belonging to a given category. The same prompts are used to classify the NSD and generated images.

