# OpenReview forum: "Brain Diffusion for Visual Exploration: Cortical Discovery using Large Scale Generative Models"
_NeurIPS.cc/2023/Conference — NeurIPS 2023 oral_

### Official Review · Reviewer_Vg6A · 2023-07-04

**Soundness:** 2 fair
**Presentation:** 3 good
**Contribution:** 3 good
**Rating:** 6
**Confidence:** 4

**Summary:**

This paper proposes a diffusion-based method for fMRI-guided image synthesis and claims that it can identify the selectivity of various ROIs in the visual cortex. The motivation of this work is well-defined and is of vital importance to the computational neuroscience field. While the method in this work is intuitive and lacks machine learning depth.

**Strengths:**

Strengths:
* Interesting scientific field and question of neuroscience.
* The paper is well-written and has a smooth and concrete flow.
* The experiments and visualizations are in good condition and sufficient.

**Weaknesses:**

Weaknesses

* The method used in this paper is heuristic and lacks technical depth.
    * Generating or reconstructing images from fMRI is not a new idea, as has been proposed in [1], etc.
    * CLIP is an off-the-shelf large-scale pre-trained model, performing conditional diffusion-based generation with embedding from CLIP or Stable Diffusion is stereotyped [2,3].
    * In Section 3.3, the domain-specific techniques you designed for this specific task are just some first-order tricks (i.e., linear combination of S, euler approximation). Maybe these tricks are effective enough, but that's overly empirical.
* The experimental section is empirical and lacks theoretical analysis. Your analysis of the fMRI encoder with ROIs is interesting, but little focus had been put on what's the scientific relationship of neuroscience and diffusion model.



[1] https://www.biorxiv.org/content/10.1101/2022.11.18.517004v3

[2] https://arxiv.org/abs/2112.10752

[3] https://arxiv.org/pdf/2208.01618.pdf

**Questions:**

Please consider the things listed in the “Weaknesses” section.

Also please consider providing information regarding the interpretability of the utilization of the diffusion model in your context.

**Limitations:**

To my knowledge and understanding, there is no potential negative societal impact of this work. Other limitations please see “Weaknesses”.

---

> ### Author Rebuttal · Authors · 2023-08-09
>
> We are strongly encouraged by your evaluation that our work is interesting for neuroscience, it is well-written, and it has good experiments. We appreciate your advice on clarification. We address specific comments below and refer to the general response for results.
>
> > **Scope of the paper**
>
> We would like to clarify that our work proposes to use diffusion models as a component to elucidate the functional role of different regions of the brain. Our work does not investigate the use of diffusion models for image reconstruction from fMRI or image generation from text. We will clarify this in the upcoming revision.
>
> We are using diffusion models because they have been demonstrated to be state of the art estimators of the natural image prior [1]. These models have been trained on billions of images (the model we use is trained on LAION-2B), which is three magnitudes larger than BigGAN models trained on ImageNet. We further use the fact that diffusion models can be interpreted as a a score function (gradient of the log-likelihood), and can be combined with a derivative of an energy function (gradient of the brain w.r.t. input) [2].
>
> > **Q1.1) Comparison to Takagi et al.**
>
> The paper by Takagi et al. [3] is relevant and interesting. It is a work we already cite (Line 67) in our own work. However the problem that we investigate is very different from Takagi et al.
>
> Takagi et al. investigates *reconstructing visual stimuli from a fixed set of voxels* in the brain. Our work explores the synthesis of **novel visual stimuli** that are predicted to activate parts of the visual cortex, not reconstruction of seen stimuli. The method proposed in our paper is intended as a tool for future data-driven investigations of the human visual cortex.
>
> As our method is gradient based, it is permutation invariant [4] and can be flexibly applied to any subset of visual cortex voxels given a voxel-wise encoding model.
>
> **To our knowledge, our paper is the among the first to explore the synthesis of images predicted to activate higher visual cortex with diffusion models.** There is concurrent work [5] which explores using diffusion models to synthesize images predicted to activate macaque V4. Previous work on diffusion models with fMRI tackle reconstruction.
>
> > **Q1.2) On text-conditioned image synthesis**
>
> To clarify, **we are not performing text-conditioned image synthesis** using CLIP embeddings as done in [6,7].
>
> Our use of CLIP is purely as a backbone for the visual encoder for fMRI. The use of an pretrained backbone with a linear decoding layer is common in voxel-wise encoding models for fMRI neuroscience, and CLIP models are the best backbone for higher visual cortex in fMRI [8].
>
> The optimization method proposed in [7] requires knowledge of ground truth images for a given concept, which we do not have.
>
> > **Q1.3) Use of linear combination of S and euler estimation**
>
> * **On a linear combination of S**
>   * A linear combination of voxels (S) activations is accepted in neuroscience investigations for fMRI. In fMRI, voxels are often clustered in regions of interest (ROIs), where voxels have similar selectivity and behavior to visual stimuli. In ROIs, it is common to evaluate the average activations of all voxels using a linear combination (average of S). **This approach has been used in some of the most important papers for visual fMRI [9, 10].**
>
> * **On an euler estimation**
>   * The euler estimate is often used when the gradients of a network are used to guide image synthesis. Concurrent work [5], which uses diffusion models to generate images that are predicted to activate macaque V4 neurons, **also uses an euler estimate to estimate a clean sample that is fed to their encoder (see e.q. 6 in [5])**. This approach is acknowledged in other work ([11] section 2.3 last paragraph; And [12] e.q. 4). We adopt this technique since the euler estimate is in the distribution of natural images, and only use it to modify the image provided to the encoder, not the diffusion output directly.
>
> > **Q2) Theoretical analysis of our work**
>
> We have included a theoretical analysis of our work in **section 9 of our supplemental.**
>
> In that section, we show that with an encoder backbone of fixed norm, the brain maximization objective for a single/multiple voxels yields an image where the CLIP image embedding is equal to the normalized voxel weight.
>
> > **Interpretability**
>
> We show in **section 4 of the supplemental** how the brain and the diffusion model work together. We find that low-level details often emerge first, and that the brain signal and diffusion signal are not always harmonious.
>
> ⠀
>
> We genuinely appreciate the reviewer's commitment to ensuring the rigor of our paper. Their insights have undeniably contributed to its refinement. The clarifications we've provided underscore the innovative aspects of our research and its potential contributions to the field. We respectfully invite the reviewer to approach our work with a more optimistic viewpoint.
>
> ⠀
>
> [1] Diffusion Models Beat GANs on Image Synthesis
>
> [2] Reduce, reuse, recycle: Compositional generation with energy-based diffusion models and mcmc
>
> [3] High-resolution image reconstruction with latent diffusion models from human brain activity
>
> [4] Deep Set Prediction Networks
>
> [5] Energy Guided Diffusion for Generating Neurally Exciting Images
>
> [6] High-Resolution Image Synthesis with Latent Diffusion Models
>
> [7] An Image is Worth One Word: Personalizing Text-to-Image Generation using Textual Inversion
>
> [8] What can 1.8 billion regressions tell us about the pressures shaping high-level visual representation in brains and machines?
>
> [9] The fusiform face area: a cortical region specialized for the perception of faces
>
> [10] A cortical representation of the local visual environment
>
> [11] GLIDE: Towards Photorealistic Image Generation and Editing with Text-Guided Diffusion Models
>
> [12] UPainting: Unified Text-to-Image Diffusion Generation with Cross-modal Guidance

---

> > ### Comment · Reviewer_Vg6A · 2023-08-19
> >
> > I thank the authors for a detailed response and for answering my concerns one by one.
> >
> > **Q1.1)** : It's true that the problem that you investigate is different from the Takagi et al. [1]. However, the method and model framework you use is much similar to them. Yours and their methods are both in the Latent Diffusion Model scope. As you mentioned in Section 3.1.
> >
> > **Q1.2)**: Thanks for your comment. This part makes sense to me. Your new technique of performing fMRI-conditioned image synthesis is interesting.
> >
> > **Interpretability**: I think the results of Section 4 of the supplementary is a characteristic of a normal diffusion model, in which it tends to generate the main features of an image in the first steps and then details in latter steps. I wonder how the gradient guidance of your method contributes to this generation process.
> >
> > In conclusion, while there's room for refinement in the methodology, the paper offers significant scientific contribution to neuroscience. I have revised my score in light of these observations.
> >
> >
> >
> > [1] High-resolution image reconstruction with latent diffusion models from human brain activity

---

> > > ### Author Response · Authors · 2023-08-20
> > >
> > > We appreciate the positive evaluation you have given to this paper!

---

### Official Review · Reviewer_SV7e · 2023-07-04

**Soundness:** 3 good
**Presentation:** 3 good
**Contribution:** 3 good
**Rating:** 7
**Confidence:** 5

**Summary:**

The paper presents a new algorithm to guide a diffusion model to decode maximally activating images for particular voxel subregions of human fMRI using an encoding model trained to predict brain activity from images. The algorithm identifies stereotypic features for defined ROIs, such as faces or food items. The authors evaluate the specificity of the reconstructed image with CLIP zero-shot classification and human evaluation. They also cluster the encoding model weights and find that the clusters result in perceivable semantic categories in the reconstructed images. This algorithm allows for more fine-grained analysis of preferences across the visual system.

**Strengths:**

There have been many papers that reconstruct images from brain activity. However, all of them require some form of retraining the diffusion model. This approach only needs an encoding model that can be trained independently.

The extensive human evaluation is great. While this goes beyond what I would ask for a NeurIPS paper, I think it would be great to also verify them by showing them back to human subjects.

**Weaknesses:**

The authors show that the reconstructed images possess stable properties that can be identified by humans and by a CLIP network. However, they don't show that the images also do what they are supposed to do, which is activate a particular subnetwork. While human experiments probably go beyond the scope of this NeurIPS paper, one way to do that would be to take **another** encoding model for the same neural activity, take it as a proxy for the brain and show the images to that.


**Questions:**

Q1: As far as I understand $\gamma$ trades off between the denoising gradient and the maximization gradient. I couldn't find what value you set it too, or how you chose it.

Q2: There is a preprint that describes pretty much the same idea as in the paper but uses it for V4 single neurons:

Pawel A. Pierzchlewicz, Konstantin F. Willeke, Arne F. Nix, Pavithra Elumalai, Kelli Restivo, Tori Shinn, Cate Nealley, Gabrielle Rodriguez, Saumil Patel, Katrin Franke, Andreas S. Tolias, Fabian H. Sinz Energy Guided Diffusion for Generating Neurally Exciting Images
https://www.biorxiv.org/content/10.1101/2023.05.18.541176v1

If one looks at the date, the preprint has been submitted around the NeurIPS deadline, so it's unlikely that the authors were aware of it and I would consider this as a case where two groups had the same idea. However, I think it would be fair to mention/cite them.

Q3: Can you discuss the compute time required for a single image?


**Limitations:**

Limitations are discussed.

---

> ### Author Rebuttal · Authors · 2023-08-09
>
> We are deeply grateful for your excellent suggestions! We will address specific questions below. Please see the PDF in the general response for additional figures.
>
> > **Additional validation**
>
> We agree with your comments on additional validation. **There is ongoing work to investigate the performance of BrainDiVE synthesized images in humans using fMRI.**
>
> Following your suggestion, we train an alternative encoding model using EVA02_CLIP_B_psz16_s8B [1], this is a very recently published backbone model with TrV blocks (in contrast to the ViT B/16 backbone with ViT blocks we use in the paper) jointly trained using mask image modeling and image-text contrastive loss (backbone we use in paper is image-text contrastive loss alone), which has been independently validated to achieve high ImageNet performance. We freeze the model weights, and train a linear probe with bias to estimate the fMRI activations. We validate that the new model can achieve high $R^2$ that is comparable to our current backbone.
>
> **Please see the PDF in the general response for a distribution plot of predicted activations.** We find that our BrainDiVE images can achieve high predicted neural activations when valdiated on a new backbone.
>
> > **Q1) Choice of $\gamma$**
>
> Indeed, a high $\gamma$ gives more weight to the brain activation gradient (more activating, less natural), while a low $\gamma$ gives more weight to the diffusion gradient (more natural, less activating). We set $\gamma$=130, and this is described in Line 149 of the original paper. There is a typo here and it was represented as $\eta$, we will update the revision to clarify this value.
>
> We performed search in increments of 10, exploring values between 10.0 up to values of 300.0, and synthesized 100 images for each of the broad category selective regions and recorded the gradient values at each step and the final synthesized images. We found that values between 110 and 150 yielded gradient magnitudes that largely matched between the brain and diffusion at early time-steps when coarse image structure emerged. The value of 130 was selected as a median value. This is the only hyperparameter that we introduce on top of diffusion models.
>
> Approaches like reflected diffusion [2] or dynamic thresholding from Imagen [3] may enable higher $\gamma$ values, and remain an avenue for future research.
>
> > **Q2) On concurrent work**
>
> We were not previously aware of the work by Pierzchlewicz et al. (2023) on "Energy Guided Diffusion for Generating Neurally Exciting Images" [4]. After closely reading the paper, we agree that the ideas in our paper and their paper are similar.
>
> Both works tackle the synthesis of images that are predicted to activate regions of the brain. A broad difference is that our paper targets the higher visual cortex in humans which are believed to encode semantic concepts, while they target V4 in macaques.  This yields differences in the final images, where they primarily target the synthesis of complex visual patterns, while we primarily target the synthesis of compositional visual scenes.
>
> We thank the reviewer for bringing this interesting paper to our attention, and will include it as a citation in our upcoming revision.
>
> > **Q3) Compute time**
>
> When using fp16 diffusion models, with fp32 brain encoders, 50 steps of denoising for images at 512x512 resolution requires around 25 seconds on a Nvidia V100 (from 2017). We briefly discuss this in section 9 of our supplemental, and will clarify this in the main paper in an upcoming revision.
>
> In practice our experiments were performed on a cluster and consumed 1500 GPU compute hours. Accelerating this processing is a very interesting avenue of research, and we discuss approaches inspired by MagicMix/SDedit [5,6] in section 4 of our supplemental to speed up our work.
>
> ⠀
>
> We hope the clarifications have been insightful. Please don't hesitate to comment if there are any additional questions!
>
> ⠀
>
> [1] EVA-CLIP: Improved Training Techniques for CLIP at Scale
>
> [2] Reflected Diffusion Models
>
> [3] Photorealistic Text-to-Image Diffusion Models with Deep Language Understanding
>
> [4] Energy Guided Diffusion for Generating Neurally Exciting Images
>
> [5] MagicMix: Semantic Mixing with Diffusion Models
>
> [6] SDEdit: Guided Image Synthesis and Editing with Stochastic Differential Equations

---

> > ### Comment · Reviewer_SV7e · 2023-08-11
> > **Thanks for the clarifications**
> >
> > Dear authors, thank you for your clarifications. I have read the rebuttal and will stand by my evaluation.

---

> > > ### Author Response · Authors · 2023-08-11
> > >
> > > We appreciate the timely response.
> > >
> > > Thank you again for the positive evaluation of this paper!

---

### Official Review · Reviewer_fpHd · 2023-07-05

**Soundness:** 3 good
**Presentation:** 3 good
**Contribution:** 3 good
**Rating:** 7
**Confidence:** 4

**Summary:**

This study proposes BrainDIVE, a system for synthesizing optimal stimuli for any given region of interest in the brain. The model combines a pretrained latent diffusion model for image generation with a linear “brain encoder” trained to map CLIP feature vectors onto the corresponding brain activity. At test time, a gradient-based optimization iteratively produces an image that maximizes the activity of a predefined set of voxels. The technique is validated on well-known ROIs, producing the expected images. It can also highlight subtle differences between ROIs selective to the same broad category, or functional subdivisions of existing ROIs. The functional hypotheses are validated by human subjects evaluating the properties of the generated pictures. The technique appears simple but can provide meaningful information for neuroscience studies.

**Strengths:**

* The method can retrieve subtle differences between ROIs selective for the same class (e.g. OFA and FFA).
* The method can highlight functional differences between sub-clusters of existing ROIS (e.g. food clusters)
* The qualitative observations are validated by behavioral evaluations from human subjects.


**Weaknesses:**

* The Related works section tends to overstate the novelty of the technique. It only mentions NeuroGen as prior work, whereas other prior studies had also attempted to generate optimal stimuli for specific ROIs: for instance, Ratan Murty et al (2021), Ozcelik et al (2022), Ozcelik et al (2023). The latter also used a diffusion model for image generation, as done here. Furthermore, you seem to be aware of at least some of these studies, since you criticized one of them in your Methods section for using a “hand-derived” prior. It would be much better to properly acknowledge all these studies in the Related works section, and clarify your method’s advantages/disadvantages (e.g. no hand-derived prior/time-consuming iterative method).

**Questions:**

* I do not understand why you systematically need to report the top-5 or top-10 out of 500 generated images? Do I understand correctly that the pool of 500 are all generated for the same objective (ROI maximization)? If yes, then what does this imply about the remaining 99% of images? Are they worse than NeuroGen? Are they actually failure cases? What do the worse 1% of images look like? If they are not representative of the expected category, does that invalidate your optimization method?
* Appendix, section 9, training objective: this section is useful to understand the image optimization process. One thing I do not understand from this section or from Figure 2 is why the optimization is performed in the diffusion latent space, rather than in CLIP space? Why do the gradients have to flow all the way into the diffusion model? Couldn’t you iteratively optimize the $CLIP_{img}$ vector for the same objective, and then use it to condition the diffusion model in one pass (as you would with a text prompt)? This should be computationally more efficient, and I don’t see any reason why it would be less accurate.


**Limitations:**

Limitations are properly acknowledged.

---

> ### Author Rebuttal · Authors · 2023-08-09
>
> We appreciate your detailed and concrete suggestions! We will incorporate all of your feedback into our paper.
>
> > **Comparisons to prior work**
>
> Indeed, our work builds upon the foundations laid by Ratan Murty et al. (2021),  Ozcelik et al. (2021), and Ozcelik et al. (2023). It was not our intention to imply otherwise.
>
> We originally included two of these papers in the "Related Works" sections of our paper (Murty on Line 35 and 72, Ozcelik on Line 67). In a revised version, we will explicitly highlight the connections between our work and these referenced papers and will additionally cite Ozcelik et al. (2021).
>
> Briefly:
> 1. Ratan Murty et al. (2021) builds upon an adversarial trained BigGAN image generator trained on imageNet with category conditioning. Their brain encoder consists of an ImageNet trained frozen ResNet50 backbone, along with a linear decoder layer.
>
>     Our work utilizes a text-conditioned diffusion image model trained on score matching with classifier-free guidance [1] (∇log[p(y|x;t)] = ∇log[p(x|y;t)]-∇log[p(x;t)]; y=text, x=image). This enables our diffusion model to sample from the unconditional image distribution via score function ∇log[p(x;t)] in the absence of text conditioning. Our brain encoder consist of a frozen ViT-B/16 backbone along with a linear probe, which enables brain conditioned image synthesis.
>
>     A major difference lies in the image synthesis process. As BigGAN is trained on category conditioning (1000 classes in a one-hot vector), Murty et al. perform a joint optimization over c (category) and z (latent). As the category is discrete, they use softmax(c\) to facilitate gradient optimization. It is not clear that a softmax is suitable, as BigGAN is not trained with non-discrete classes. An alternative could be gumbel-softmax [1], but this has high gradient variance.
>
>     In contrast, our work performs end-to-end differentiable optimization. So BrainDiVE is not restricted to a particular ImageNet class like NeuroGen by Gu et al., or convex combination of class embeddings as in Murty et al.
>
> 2. The method by Ozcelik et al. (2023) is state of the art for brain based image reconstruction using diffusion models, and is conditioned on a pattern of brain activations.
>
>     In contrast, the primary goal of our work is to generate images that are predicted to maximize a given region of the brain, and not reconstruction. In BrainDiVE, we only need to train an fMRI encoder, and the region can be flexibly defined as any subset of the brain without retraining.
>
>    In Ozcelik et al., when they perform region activation, they set activated voxels to 1, and others to 0. This is followed by manual latent normalization. They achieve intriguing results using this hand-crafted heuristic, but have the implicit assumption that other voxels are zeroed when one region is active.
>
> > **Q1): On top-5 or top-10**
>
> To clarify, the top-5 and top-10 are for qualitative **visualization only**. Our numerical evaluation follows prior work in image generative models (DALL-E [2]) and brain based activating image synthesis (NeuroGen by Gu et al. 2022, citation 9 in paper), and uses top-20% like NeuroGen unless otherwise noted.
>
> Our top-5/top-10 for visualization is done automatically using the brain encoder, without any manual cherry picking.
>
> For numerical results, we use the brain encoder without manual cherry picking to evaluate the top 10% and 20% (100 or 200 images) in 4.2, and follow NeuroGen and use the top 20% in 4.3/4.4. Notably this is used for visualization *and* evaluation in OpenAI's DALL-E (called reranking) `all samples used for both qualitative and quantitative results ... use reranking with N = 512` [2] where they use `top 32 of 512 after reranking with CLIP, but we do not use any manual cherry-picking` (online post). Like DALL-E, reranking is used in NeuroGen (Gu et al. 2022) where they only analyze the top-100 of 500 images as reranked using their brain encoder (confirmed with author), and only visualize the top-k of 500 images. We also rerank using our brain encoder.
>
> In general, diffusion models do not always converge during generation. The bottom 1% for the brain generally look like the bottom 1% for text-conditioned, and usually have no recognizable objects. This is an active area of research (see Imagen's dynamic thresholding approach).
>
> > **Q2): Optimizing CLIP latent**
>
> This is indeed an idea we had as well! We believe there are broadly two ideas you are touching upon:
>
> 1. Given that we know the optimal fixed CLIP image embedding (512 or 768 dim), in theory it should be easy to condition the diffusion model on this embedding to replace the typical CLIP text embedding.
>
>     In practice high performance diffusion models are text-conditioned, but are not conditioned on a fixed sized (512/768) text embedding, and instead are conditioned on 75 token embeddings (plus start/end tokens) for an entire sentence in order to model visual semantic composition.
>
>     Solving for a fixed sized CLIP image embedding does not give us the full text embedding. In addition, there exists a modality gap between CLIP image and text latents [3], and closing the gap is an active area of research.
>
> 2. An alternative may be to solve for the CLIP text embedding directly via gradient descent. However current work [4] requires a handcrafted text prompt, and has been demonstrated only for single objects. In addition, these approaches require ground truth images for a given concept, which we do not have.
>
> ⠀
>
> Thank you again for your wonderful suggestions, and we're eager to hear your thoughts on our clarifications. Please let us know if you have any additional questions or comments!
>
> ⠀
>
> [1] Categorical Reparameterization with Gumbel-Softmax
>
> [2] Zero-Shot Text-to-Image Generation
>
> [3] Mind the Gap: Understanding the Modality Gap in Multi-modal Contrastive Representation Learning
>
> [4] An Image is Worth One Word: Personalizing Text-to-Image Generation using Textual Inversion

---

> > ### Comment · Reviewer_fpHd · 2023-08-16
> >
> > Thank you for your responses, which I found clear and satisfactory. In consequence, I have raised my score to 7.

---

> > > ### Author Response · Authors · 2023-08-16
> > >
> > > We are deeply grateful for the positive assessment you've given to our paper! Thank you again for your suggestions and comments.

---

### Official Review · Reviewer_MC3H · 2023-07-21

**Soundness:** 3 good
**Presentation:** 3 good
**Contribution:** 3 good
**Rating:** 7
**Confidence:** 4

**Summary:**

This paper introduces Brain Diffusion for Visual Exploration (BrainDiVE) that aimed at exploring the fine-grained functional organization of the human visual cortex. Motivated by the limitations of previous studies that relied on researcher-crafted stimuli, BrainDiVE leverages generative deep learning models trained on large-scale image datasets and brain activation data from fMRI recordings. The proposed method uses brain maps as guidance to synthesize diverse and realistic images, enabling data-driven exploration of semantic preferences across visual cortical regions. By applying BrainDiVE to category-selective voxels and individual ROIs, the authors demonstrate its ability to capture semantic selectivity and identify subtle differences in response properties within specific brain networks. It is also shown that BrainDiVE identifies novel functional subdivisions within existing ROIs, highlighting its potential for providing new insights into the human visual system's functional properties.

**Strengths:**

The paper's methodology shows promise in capturing semantic selectivity and identifying fine-grained functional distinctions within visual cortical regions. Also, the paper's potential significance lies in applying BrainDiVE to understand the fine-grained functional organisation of the human visual cortex.
By providing insights into category selectivity, response properties, and sub-regional divisions, the paper opens avenues for further exploratory neuroscience studies.
To achieve this, the authors perform the experiments in the manuscript are extensive, covering:
1. the semantic specificity of the method by decoding images from task fMRI and literature-obtained ROIs
2. compared the abstraction of face representation in the brain by comparing images decoded from two regions, the fusiform face area (FFA) and the occipital face area (OFA).
3. Use the method to extend knowledge of brain function by finding subdivisions in known areas in the cases of food decoding.
The authors describe the experimental setting clearly at the beginning of section 4 for all cases.
The experiments show that the manuscript has a good balance between the combination of known methodological approaches and addressing interesting questions in neuroscience. Specifically, the authors perform a commendable effort in obtaining quantitative results using human evaluators (cf Tables 3 and 4).
Finally, the authors show through results evidence that their method is a window into understanding region-specificity hypotheses of brain regions which are knowingly involved in different aspects of visual processing.

**Weaknesses:**

The paper could benefit from comparing its results with previous contributions [e.g. 9 and 10] to highlight its novelty and contributions in relation to existing methods.
The methodology part lacks clarity, particularly in explaining key components like the diffusion model architecture and brain-guided synthesis—further clarifying the role of the image-to-brain encoder in influencing the denoising process during inference.
The generalisability and reliability of results are hard to asses through a small dataset, specifically just 10 subjects.
Furthermore, the paper lacks clarity on how the sub-divisions of the visual cortex are being verified or validated. Whether the results are specific to the analysed regions or the algorithm is being too biased by the experimental condition is not clear through the experiments. For instance, what would happen if the authors try to decode an area not specific to visual processing? An example of this would be using subsections of the orbitofrontal gyrus or other brain areas not expected to perform well in reconstructing visual stimuli.

As a small point authors should review the presentation of images. Figure 1 shows mostly best-case scenarios which are then not as good in Figure 4 (for instance the case of images generated from face voxels). This might bias readers. Second, some details, such as figure 4 appearing before figure 3 make reading the paper confusing.

**Questions:**

1. Validation of BrainDiVE's Effectiveness:
  a. Can the authors provide more details on the validation process to ensure that BrainDiVE-generated images indeed effectively activate the targeted brain regions?
  b. Could the authors consider conducting more extensive comparisons with other existing methods [e.g. 9 and 10], to demonstrate the advantages and uniqueness of BrainDiVE in eliciting specific brain activations?
  c. Could the authors show the results on decoding a baseline region which is not involved in visual processing?

2. Clarity in Methodology: The methodology part could benefit from more clarity and detailed explanations of key components, such as the diffusion model architecture and the exact implementation of brain-guided synthesis. Improving the architecture in Figure-2 might help.

3. Statistical Analysis of Qualitative Evaluation: As the paper relies on qualitative evaluation with 10 subjects, could the authors mention this explicitly and what limitations are expected from this in the limitations section?

4. Ethical Considerations: Could the authors describe the demographics of the population, or acknowledge the lack of this information to inform of wether the results are biased to a specific gender or population group.

5. Reproducibility: Are the authors intending to release the code and not publicly available data such as the scores produced by the human raters upon acceptance of the paper? This is key to guarantee reproducibility

**Limitations:**

Authors have not addressed the negative societal impact of their work, but this can be fixed by adding specific text in the Discussion section. The authors don't mention the demographic characteristics of the small human subject database that they have used not mention any ethical concerns related to decoding images from brain activations. This is fixable so authors should address it.

---

> ### Author Rebuttal · Authors · 2023-08-09
>
> Thank you for the excellent suggestions. We address specific questions below, and will include additional details in the general response.
> > **Formatting and presentation**
>
> We will update our paper to follow other image synthesis papers like OpenAI's DALL-E 2 [1] and GigaGAN [2], they similarly select/curate images for the first figure and note this in the caption. We will also update the figure numbering.
>
> > **Q1.1) Validation of effectiveness**
>
> For within ROI clusters, OPA & food clusters remained stable across random cluster initializations, and showed the highest cosine distance for voxel embeddings among all ROIs we examine.
>
> More broadly, we agree on the importance of validating and conducting fMRI studies to evaluate BrainDiVE. **We have ongoing work in this area**.
>
> In the paper we perform validation via two methods:
> 1. When we target voxels from widely accepted category selective regions (faces, places, bodies, food, word), we perform CLIP 5-way classification using a natural language probe. The probe sets follow the guidelines detailed in section 3.1.4 of [4]. We provide our full probe set in page 26 of our supplemental.
>
>     We found that images created using BrainDiVE were aligned with scientific consensus for broad category selective visual regions.
>
> 2. When voxels from single ROIs (FFA/OFA), and within ROIs (Food/OPA) are targeted, we perform behavioral studies. Ten different subjects were recruited for each region (FFA, OFA, Food, OPA) via Prolific. This yielded 80 evaluations per-subject per-question, for 10 subjects each. We find that BrainDiVE can highlight differences between ROIs (FFA vs OPA) and within ROIs (Color in food voxels, and indoor/outdoor in OPA). This is important as it may facilitate future open-ended data driven exploration of the visual cortex.
>
> We further validate using an alternative fMRI encoder backbone in the **response PDF**, and find our images can achieve high predicted activations.
> > **Q1.2) Comparisons with existing methods**
>
> We concur that our work builds upon Gu et al. and Ponce et al. ([9, 10] in text). In particular, the setup of our work is similar to [9], as we both use the NSD dataset and operate offline (no live subject). Our work is not directly comparable to [10] as their experiments use real-time macaque brain feedback, which our work and [9] do not have.
>
> Using BrainDiVE, we probe the visual cortex selectivity of the brain at three hierarchical stages, and offer new insight on region wise selectivity. Briefly, our work differs from [9] in two important ways:
>
> 1. We use a diffusion model trained on billions of internet images, three magnitudes larger than BigGAN (ImageNet), this is important as we are not restricted to single object images that dominate ImageNet. We further leverage the model's training with classifier-free guidance [5] (∇log[p(y|x;t)] = ∇log[p(x|y;t)]-∇log[p(x;t)], with y=text, and x=image) to use the model in an unconditional mode, and combine it with our brain encoder to enable brain conditioned sampling. This is in contrast to Gu et al., where they use BigGAN trained on imageNet and condition on class-labels.
>
> 2. The brain optimized image generation process is very different. [9] uses search then optimize. They first sample images the 1000 ImageNet classes, and **non-differentiably** select the top-10 classes for each region. Finally they perform brain gradient updates of the GAN latent. This results in images with less diversity, and many of their images are nearly identical. In contrast, our work yields diverse images by using end-to-end differentiable optimization, **and do not restrict the search space to a fixed image category a priori**.
>
> Please see the **response PDF** for more comparisons.
>
> > **Q1.3) Non-visual cortex results**
>
> Please see **response PDF** for OFC results. As expected, we find our method does not yield consistent semantics in OFC.
>
> > **Q2) Clarity in methodology**
>
> We agree that this could be further clarified. We will provide an updated Figure 2 in an upcoming revision.
>
> The brain encoder predicts fMRI activations from images. Diffusion models predict the score (derivative of log-likelihood) of the image distribution, and can be treated as a special class of energy-based models. A brain encoder that outputs real-numbers can be interpreted as an energy function [6], the derivative of which can be additively combined with the diffusion to enable conditional sampling of naturalistic images that maximize brain response.
>
> > **Q3) Statistical analysis of human evaluation**
>
> Each of the 10 subject performed 80 binary evaluations per question, we collect 8800 total responses (11 questions, 10 splits, 4 NSD subjects, for both BrainDiVE/real images). Due to space constraints, we provided standard error (SE) measurements in section 5 of the supplemental. We will provide further statistical analysis in an upcoming revision.
>
> > **Q4) Ethical Considerations**
>
> For 30 subjects, they are between age 22~63, 14 women/15 men/1 unknown, 28 white, 2 black, 1 mixed, 1 unknown. We currently have a "Broader Impacts" section in the supplemental, we will update this in the revision to discuss the demographics and additional issues.
>
> > **Q5) Code release**
>
> Yes, we will release code and human evaluation data upon acceptance. We have also sent the AC a comment linking to an anonymous repo containing our code.
>
> ⠀
>
> We're grateful for your clear and helpful suggestions. In light of our response, we hope you might view our work in a more positive light. Please feel free to let us know if you have additional comments!
>
> ⠀
>
> [1] Hierarchical Text-Conditional Image Generation with CLIP Latents
>
> [2] Scaling up gans for text-to-image synthesis
>
> [3] Selectivity for food in human ventral visual cortex
>
> [4] Learning transferable visual models from natural language supervision
>
> [5] Classifier-free diffusion guidance
>
> [6] Reduce, reuse, recycle: Compositional generation with energy-based diffusion models and mcmc

---

> > ### Comment · Reviewer_MC3H · 2023-08-19
> >
> > Thanks for the responses. I have updated my score to “Accept” following the suggestions for the evaluation of a good contribution with impact in a specific field

---

> > > ### Author Response · Authors · 2023-08-19
> > >
> > > We are very grateful for the positive assessment you've given to our work!
> > >
> > > We would like to again express our appreciation for your suggestions.

---

> ### Author Response · Authors · 2023-08-14
>
> We appreciate the request for code. The anonymous github for the paper is now in a publicly visible comment.
>
> Also reproduced here for convenience: https://anonymous.4open.science/r/BrainDiVE_code-FF0C

---

### Author Rebuttal · Authors · 2023-08-09

We are grateful to all reviewers for their constructive suggestions, which we agree will significantly improve the communication of our work.

We are very encouraged by reviewers’ evaluation on the quality of this paper. All four reviewers find the work interesting ("methodology shows promise in capturing semantic selectivity" (MC3H); "The method can retrieve subtle differences between ROIs" (fpHd); compared to prior works -- "this approach only needs an encoding model that can be trained independently" (SV7e); "The experiments and visualizations are in good condition and sufficient" (Vg6A)).


### General clarifications
### 1. Scope and experiments

* We propose BrainDiVE -- a method that utilizes diffusion models to investigate functional specialization in the higher visual cortex.
* Our work relies on end-to-end differentiable optimization with an image-computable voxel-wise fMRI encoder, and leverages an existing large-scale image diffusion model without retraining. It can be flexibly targeted at arbitrary regions in the visual cortex without retraining.
* We apply BrainDiVE to explore the brain's visual cortex selectivity at three hierarchical levels and offer new scientific insight on the selectivity of different regions.
    * First, we apply it to widely accepted category selective regions -- faces, places, bodies, food, words.
    * Second, we apply it to ROIs that code for faces at different levels in the feature hierarchy (FFA, OFA). Our results are in line with the widely held belief that OFA responds to lower level face features relative to FFA.
    * Third, we apply it to splits within food-selective and place-selective (OPA) ROIs, where we identify potentially novel functional subdivisions.
* Our evaluation is done in two different ways
    * For category selective regions, we perform CLIP 5-way classification using natural language. The design of our prompts follow the best practices defined in 3.1.4 of [1]. The prompts are available in section 9 of the supplemental. We observe that BrainDiVE images indeed capture the category specificity of these regions.
    * For FFA and OFA, which both code for face; subregions of food; and subregions of OPA which codes for scenes -- we perform a human behavioral study to validate the fine-grained visual-semantic attributes. We recruit 10 subjects for each set of comparisons. This results in a total of 8800 total responses (11 questions, 10 splits, 4 NSD subjects, for both BrainDiVE and real images). We report standard error metrics in section 5 of the supplemental. We find that BrainDiVE can highlight differences in preferred attributes between visual regions, suggesting that it can be useful for future data driven exploration of the visual cortex.

The ROI masks are derived from functional localizer results from the official NSD paper (faces, places, bodies, words, OFA, OPA), or obtained from other authors directly (food) [2].

We include visualizations of the image synthesis process and brain gradients in **section 4** of the supplemental. We perform a theoretical evaluation of our work in **section 9** of the supplemental.

We agree in the importance of evaluating the images using real humans, and an effort to collect fMRI data is ongoing.

### 2. Relationship to concurrent and prior work

To our knowledge, we are the first to apply diffusion models to investigate the selectivity of the human higher visual cortex. Unlike prior work, we are not doing image reconstruction. There is concurrent work by Pierzchlewicz et al. [3] which applies diffusion models to investigate the selectivity of macaque V4, we will cite this in an upcoming revision.

In our work we cite papers which we believe are highly relevant (in-paper citation numbers): Gu et al. [9], Ponce et al. [10], Murty et al. [11], Takagi et al. [46], Ozcelik et al. [47]. We build upon the insights from these works. Our work is most similar to [9], as we use the same dataset, while [10] uses real-time responses of macaque visual cortex neurons for image synthesis.

* Unlike [9, 11], we perform end-to-end gradient optimization, and do not rely on fixed categories identified via search as in [9], or a softmax relaxation of categories as in [11]. In addition, we use a diffusion model trained on billions of images. Due to this, our synthesized images are not restricted to single object images that dominate the 1000 classes from ImageNet as in [9, 11].
* Different from [46, 47], we are not performing reconstruction of visual stimuli. Instead, we are proposing to use a diffusion model as a component for the synthesis of novel visual stimuli that is predicted to match the selectivity of a region. We do not rely on a fixed sized input, and regions can be flexibly defined without retraining. [47] achieves intriguing results by setting voxels to 0/1 followed by latent normalization, but this relies on the implicit assumption that voxels outside a region are inactive.

We will include a more extensive discussion of related work in an upcoming revision.

### 3. New figures/results in response PDF
1. We further validate the BrainDiVE results using an encoder with an alternative pretrained backbone. We find that the results are robust.
2. There is a more extensive comparison of BrainDiVE and NeuroGen from Gu et al. This highlights the semantic fidelity and diversity of our images.
3. We apply BrainDiVE to a region outside the visual cortex, which is not known to have visual selectivity. We confirm that the images do not show consistent semantic trends.

Additional results will be included in an upcoming revision.

⠀

We genuinely appreciate the suggestions, and believe our paper will be improved with your feedback. Please let us know if you have any additional questions or comments!

⠀

[1] Learning transferable visual models from natural language supervision

[2] Selectivity for food in human ventral visual cortex

[3] Energy Guided Diffusion for Generating Neurally Exciting Images

---

### Author Response · Authors · 2023-08-09
**Link to code**

Here is a copy to a fully anonymized version of the core code used in this paper:
https://anonymous.4open.science/r/BrainDiVE_code-FF0C

The code includes the following:
* fMRI encoder architecture
* fMRI encoder training
* fMRI encoder validation
* Diffusion architechture
* Diffusion inference using broad category selective regions; ROIs; and parts of ROIs

We are fully committed to releasing a complete version of the code at a later point in time.

---

### Decision · Program_Chairs · 2023-09-21

**Decision:**

Accept (oral)

**Comment:**

The paper presents a novel approach to reconstruct images from brain activity that only needs an encoding model that can be trained independently.
The main interest is that uses the most recent developments on generative models of vision to perform novel analyses of the visual cortex in humans and obtain non-trivial results on the semantic specialization of visual regions.
The method can indeed retrieve subtle differences between ROIs selective for the same class (e.g. OFA and FFA) and highlight functional differences between sub-clusters of existing ROIS.
The qualitative observations are validated by behavioral evaluations from human subjects.

The paper is well-written, and the experiments and visualizations are in good condition and sufficient.
On the weakness side, the technical contribution of the paper is overall less impressive.

There is a wide consensus among reviewers, hence the paper is certainly well suited for NeurIPS, and certainly for a spotlight presentation.